# Prevalence of intestinal parasites in school-age children in Turkey: A systematic review and meta-analysis

Ahmed Galip Halidi[1]*, Kemal Yaran[1], Selahattin Aydemir[2], Abdurrahman Ekici[2], Yusuf Dilbilir[3]

1 Vocational School of Health Services, Muş Alparslan University, Muş, Turkey, 2 Faculty of Medicine, Van Yüzüncü Yıl University, Van, Turkey, 3 Vocational School of Health Services, Hakkari University, Hakkari, Turkey

* g.sarisu@alparslan.edu.tr

## Abstract

### Background

Intestinal parasites are a major public health problem worldwide, especially in societies with low socioeconomic status and where sanitation rules are not sufficiently emphasized. School-age children are the most affected group in these societies. In light of the literature data, the intestinal parasites detected in school-age children and the geographical and socioeconomic structure of Turkey are evaluated together. The study hypothesizes that the parasite prevalence in school-age children is high, and there is substantial socioeconomic and geographical variation in species-specific prevalence. It is aimed to determine the pooled prevalence of intestinal parasite infections in school-age children in Turkey, identify the common parasite species, and compare the prevalence in different geographical regions of Turkey.

### Methods

The Web of Science, PubMed, Scopus, and TR index databases were searched to access published articles reporting the presence of intestinal parasites among school-age children in Turkey. The prevalence of intestinal parasites was calculated using a random effects model. Subgroup analyses were performed according to the parasite species and geographical regions of Turkey. Also, year-based meta-regression analyses were conducted.

### Results

A total of 204.754 samples from 99 articles were included in the analysis. The pooled prevalence of intestinal parasites was 29%, with high heterogeneity ($I^2 = 99.88\%$, $P < 0.001$). The subgroup analysis revealed that the Southeastern Anatolia is the region with higher prevalence of intestinal parasites among school-age children,

**Data availability statement:** All data are in the manuscript and/or Supporting information files.

**Funding:** The author(s) received no specific funding for this work.

**Competing interests:** The authors have declared that no competing interests exist.

with a rate of 41% ($I^2 = 99.44\%$, $P < 0.001$). Subgroup analysis for parasite species revealed that *G. intestinalis/duedonalis/labmblia* (11%; 95% CI: 9%-13%, $I^2 = 99.85$) was the most frequently detected parasite in school-age children, followed by *Blastocystis spp., Enterobius vermicularis, Entamoeba coli, Ascaris lumbricoides* and *Entamoeba histolytica/dispar*.

## Conclusion

The prevalence of intestinal parasitic infections (IPI) among school-age children in Turkey is particularly high in Southeastern Anatolia, the Mediterranean, and Eastern Anatolia. Socioeconomic conditions, education, and geography are the main factors that affect this situation. It is crucial for school-age children and their parents to receive education on the transmission mechanisms of intestinal parasitic infections (IPIs) and strategies for their prevention. Furthermore, it is essential for local governments and public authorities to upgrade infrastructure to ensure that drinking water and food are not contaminated by polluted water sources.

### Author summary

Intestinal parasite burden consisting of helminths and protozoa causes serious public health problems worldwide. Studies conducted in Turkey indicate that this phenomenon is particularly prevalent among school-aged children. The primary motivation for this meta-analysis was the imperative to tackle this problem from a holistic perspective. Turkey consists of seven regions and 81 provinces. In a country with 86 million inhabitants, it is obvious that bringing together studies on the subject in such a study and sharing their problems with the scientific community and the public will raise awareness in the fight against intestinal parasites. In this study, the prevalence of IPI was 29% ($I^2 = 99.88$, $P < 0.001$). When the regional subgroup analysis results were analyzed, it was 41% ($I^2 = 99.44$, $P < 0.001$) in the Southeastern Anatolia region. A second subgroup, i.e., parasites, was examined separately and it was found that *G. intestinalis/duedonalis/labmblia* (11%; 95% CI: 9%-13%, $I^2 = 99.85$) was the most frequently detected parasite in school-age children, followed by *Blastocystis spp., Enterobius vermicularis, Entamoeba coli, Ascaris lumbricoides* and *Entamoeba histolytica/dispar*. The prevalence of IPI among school-age children is high in Turkey, especially in the Southeastern Anatolia, Mediterranean, and Eastern Anatolia regions. Socioeconomic conditions, education, and geography are the main factors contributing to this situation. It is of utmost importance that local authorities and the State initiate awareness raising and awareness training on IPI among school-age children and their families, focusing on mechanisms to prevent the transmission of parasitic agents.

## Introduction

Intestinal parasites are recognized as a major public health problem worldwide. Based on existing studies, approximately 3.5 billion individuals are infected with some form of intestinal parasites [1,2]. Among intestinal parasites, microscopic single-celled protozoa and cm-long helminths are found. Understanding the detailed life cycle of these parasites is crucial for effective combatting. The golden rule in the fight against all parasites is to interrupt the parasite's life cycle at some point in time. This is sometimes possible in the final host and, at other times, in the intermediate host [3,4]. Lack of sanitation, unhygienic conditions, and inadequate health education in underdeveloped or developing countries are major reasons for the rapid increase in these parasitic infections in communities. IPIs have been implicated as a major cause of digestive disorders (such as diarrhea, nausea, and vomiting), chronic malabsorption and malnutrition, and failure to thrive, especially among high-risk groups such as children, pregnant women, and immunocompromised patients [5,6].

School-age children are particularly susceptible to IPIs because of their prolonged exposure to shared environments, such as common restrooms and playgrounds, coupled with a lack of established hygiene and sanitation practices. IPIs, which are frequently seen in school-age children, not only affect children's health but also negatively affect their educational life and learning processes. For these reasons, IPIs in children are a serious health problem [3,7,8].

Analysis of existing studies indicates that the prevalence of intestinal parasites among school-aged children in Turkey is significantly high, a condition influenced by the country's geographical and socioeconomic factors. This represents a substantial public health issue that is frequently neglected. Thus, this study aimed to determine the pooled prevalence of intestinal parasitic infections in school-age children in Turkey, identify the common parasite species, and compare the prevalence in different geographical regions of Turkey.

## Methods

This study systematically reviewed and analyzed published articles using a meta-analysis approach to estimate the prevalence of intestinal parasites among school children in Turkey. The literature search, publication selection, and results were conducted according to the PRISMA guidelines (S1 Checklist) [9]. The protocol for this systematic review and meta-analysis was registered in the International Prospective Register of Systematic Reviews (PROSPERO) database. The registration number is CRD42024576183, and the protocol.

### Search strategy

A comprehensive literature search was performed using all identified keywords in four electronic databases (PubMed, Web of Science, TR index, and Scopus) on 20.08.2024, to identify relevant studies that reported the prevalence of intestinal parasites among school children in Turkey. This study did not impose restrictions on the year or language features. Moreover, a manual search was conducted using references from the retrieved articles to identify additional relevant studies that might have been missed. The detailed search strategy for all the databases is presented in S1 Table.

### Data management and study selection

All identified articles were initially retrieved and managed using Mendeley software. After the removal of duplicates, relevant studies were independently selected by two authors (AGH and KY). The titles and abstracts of the retrieved studies were evaluated based on the eligibility criteria. Subsequently, articles with any potential for inclusion or any uncertainty about eligibility were subjected to a full-text review. Any disagreement or uncertainty was resolved by discussion and, when necessary, by a third reviewer (SA and AE). Furthermore, attempts were made to obtain missing data or clarify any uncertainty with the corresponding authors. Articles reporting the same research data/findings published in different formats/titles by the same author were counted only once.

## Inclusion and exclusion criteria

The eligibility of full-text articles for inclusion in this study was evaluated using the following inclusion criteria: (1) cross-sectional studies; (2) conducted in Turkey and reporting the prevalence of intestinal parasites; and (3) studies published before August 2024. The exclusion criteria were as follows: (1) case reports, reviews, and studies without original data; (2) non-cross-sectional studies; (3) overall prevalence not reported and impossible to estimate based on the results and confusing or unclear analysis results; (4) surveys conducted in a hospital or healthcare facility; and (5) articles that had limited access and those of authors who did not respond to emails twice.

## Definition of intestinal protozoan infection and outcome measures

In the context of this study, an IPIs were defined as detection of one or more of the following, intestinal parasites: *G.intestinalis/duedonalis/labmblia, Enterobius vermicularis, Blastocystis spp, Entamoeba coli, Ascaris lumbricoides, Other, E. histolytica/dispar, Hymenolepis nana Trichurus trichura,Taenia spp., Iodamoeba butschlii, Cryptosporidium spp., Dientamoeba fragilis, Entamoeba spp., Endolimax nana, Chilomastix mesnili, Fasciola hepatica,Microsporidia spp., Entamoeba hartmanni, Trichinella spp, Trichomonas intestinalis, Enteromonas hominis Trichomonas hominis, Toxoplasma gondii, Cyclospora cayetanensis, Retortamonas intestinalis, Strongyloides stercoralis, Dicrocoelium dendriticum, Hookworms, Cystoisospora belli, Balantidium coli, Isospora belli, Pentatrichomonas hominis,* The main outcome of this systematic review and meta-analysis was the estimated pooled prevalence of IPIs among school children in Turkey. The prevalence of IPIs was defined as the proportion of positive samples to the total number of samples tested.

## Data extraction

Relevant data from each eligible article were extracted and entered into a predefined Excel spreadsheet by the two authors (AGH and KY). Before the inclusion of data in the review, the extracted information was checked twice by SA, YD, and AE to ensure consistency and the absence of bias and to minimize errors. The following data were extracted: first author's name, year of publication, country and region where the study was conducted, sample size, total number of cases, identified species, and number of identified species.

## Quality assessment

The methodological quality of each included study was appraised by two independent authors (AGH and KY) using the Joanna Briggs Institute (JBI) for prevalence studies, which has nine checklist items with four options: "yes", "unclear" and "not applicable" [1,10,11].

## Data analysis

The prevalence estimates and corresponding 95% confidence intervals (CI) were calculated for each study. The prevalence data were then pooled through a statistical meta-analysis using the restricted maximum likelihood (REML) method for the random-effects model. A forest plot was generated to summarize the results and heterogeneity among the included studies. Heterogeneity among studies was assessed using I2 statistics, in which $I^2$ values greater than 75% indicated substantial heterogeneity [1,12]. The significance of heterogeneity was assessed using Cochran's Q-test. Publication bias was assessed visually using a funnel plot and objectively using Egger's regression test. Data analysis was performed, and a plot was created using STATA 18.

## Results

### Study selection

A total of 704 articles were initially identified in the four databases. After 148 duplicates were removed, another 414 studies were excluded from the remaining articles after title and/or abstract evaluation. Furthermore, 43 articles were excluded

during the full-text assessment (S2 Table). Finally, only 99 (14%) articles met the eligibility criteria and were included in the systematic review and meta-analysis (Fig 1).

## Characteristics of included studies

The detailed characteristics of the included studies are summarized in Table 1. The 99 eligible studies were conducted in 41 provinces across seven regions of Turkey. Central Anatolia had the highest number of eligible studies (28 studies),

**PRISMA 2020 flow diagram for new systematic reviews which included searches of databases and registers only**

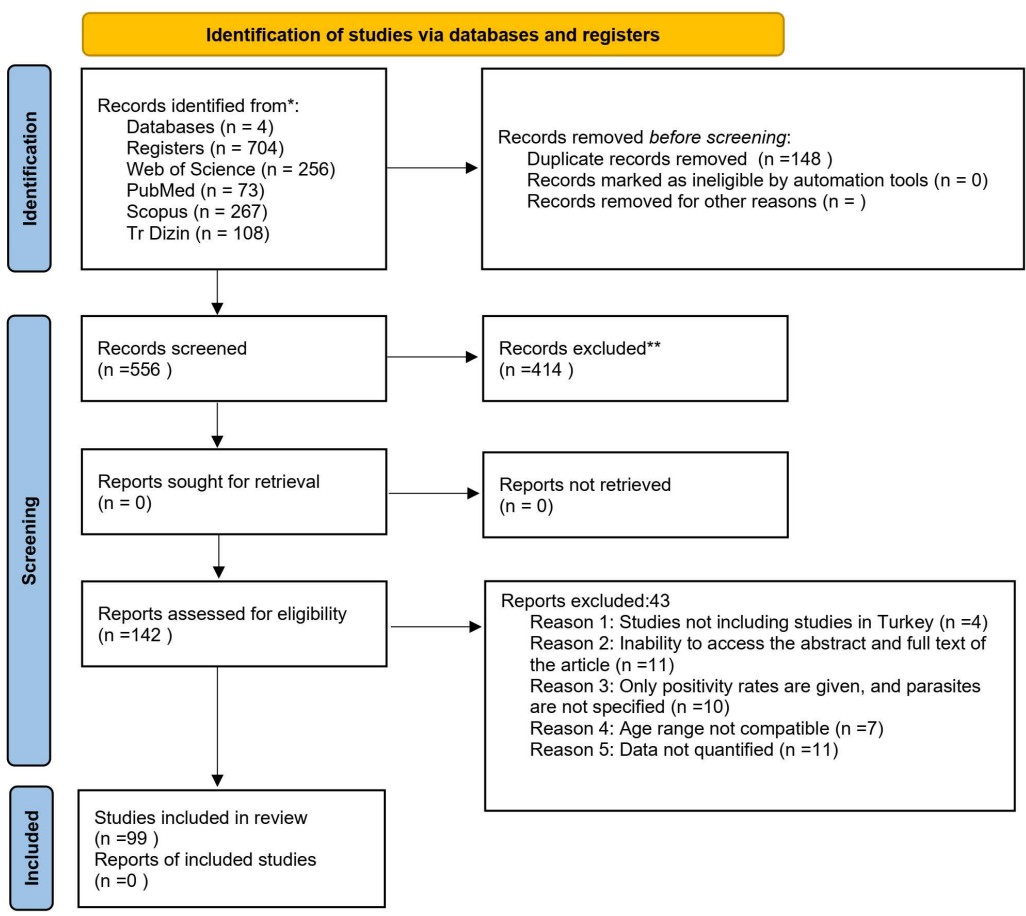

*Consider, if feasible to do so, reporting the number of records identified from each database or register searched (rather than the total number across all databases/registers).

**If automation tools were used, indicate how many records were excluded by a human and how many were excluded by automation tools.

Source: Page MJ, et al. BMJ 2021;372:n71. doi: 10.1136/bmj.n71.

**Fig 1. PRISMA 2020 flow diagram of study selection.**

**Table 1. General characteristics of the studies included in the systematic review and meta-analysis.**

| No | First Autor. (Year) | Ref. no | Region | Province | Sample Size | Positive | Prevalence With 95% CI | 95% Lower CI | 95% Upper CI | Weight (%) | Std. err. for Proportion |
|---|---|---|---|---|---|---|---|---|---|---|---|
| 1 | Aksoy, Ü., et al. (2007) | [13] | Aegean | İzmir | 1127 | 337 | 0,30 | 0,27 | 0,33 | 1,03 | 0,0136 |
| 2 | Tüzemen, NÜ., et al. (2017) | [14] | Marmara | Bursa | 9835 | 356 | 0,04 | 0,03 | 0,04 | 1,03 | 0,0019 |
| 3 | Ulukanligil, M., et al. (2004) | [15] | Southeastern Anatolia | Şanlıurfa | 908 | 500 | 0,55 | 0,52 | 0,58 | 1,03 | 0,0165 |
| 4 | Vezir, S., at al. (2019) | [16] | Central Anatolia | Ankara | 185 | 33 | 0,18 | 0,12 | 0,23 | 1,01 | 0,0281 |
| 5 | Çeliksöz, A., et al. (2016) | [17] | Central Anatolia | Sivas | 2144 | 900 | 0,42 | 0,40 | 0,44 | 1,03 | 0,0107 |
| 6 | Özkalp, B., et al. (2010) | [18] | Central Anatolia | Niğde | 110 | 41 | 0,37 | 0,28 | 0,46 | 0,97 | 0,0461 |
| 7 | Yılmaz, EA., et al. (2016) | [19] | Central Anatolia | Ankara | 211 | 21 | 0,10 | 0,06 | 0,14 | 1,02 | 0,0206 |
| 8 | Simsek, Z., et al. (2004) | [20] | Southeastern Anatolia | Şanlıurfa | 160 | 80 | 0,50 | 0,42 | 0,58 | 0,99 | 0,0395 |
| 9 | Yetkin, A., et al. (2010) | [21] | East. Anatolia | Van | 110 | 34 | 0,31 | 0,22 | 0,40 | 0,98 | 0,0441 |
| 10 | Ak, M., et al. (2006) | [22] | Southeastern Anatolia | GAP* | 4470 | 1852 | 0,41 | 0,40 | 0,43 | 1,03 | 0,0074 |
| 11 | Dagci, H., et al. (2008) | [23] | Aegean | İzmir | 2047 | 566 | 0,28 | 0,26 | 0,30 | 1,03 | 0,0099 |
| 12 | Maçin, S., et al. (2017) | [24] | Southeastern Anatolia | Şırnak | 319 | 133 | 0,42 | 0,36 | 0,47 | 1,01 | 0,0276 |
| 13 | Sankur, F., et al. (2017) | [25] | Aegean | Muğla | 468 | 35 | 0,08 | 0,05 | 0,10 | 1,03 | 0,0122 |
| 14 | Aksoy, Ü., et al. (2003) | [26] | Aegean | İzmir | 142 | 32 | 0,23 | 0,16 | 0,29 | 1,00 | 0,0351 |
| 15 | Miman, Ö., et al. (2018) | [27] | East. Anatolia | Malatya | 63 | 21 | 0,33 | 0,22 | 0,45 | 0,94 | 0,0594 |
| 16 | Ekici, A., et al. (2021) | [28] | East. Anatolia | Van | 150 | 43 | 0,29 | 0,21 | 0,36 | 0,99 | 0,0369 |
| 17 | Özdil, K., et al. (2020) | [29] | Central Anatolia | Nevşehir | 663 | 38 | 0,06 | 0,04 | 0,08 | 1,03 | 0,0090 |
| 18 | Bahceciler, NN., et al. (2007) | [30] | Marmara | İstanbul | 997 | 490 | 0,49 | 0,46 | 0,52 | 1,03 | 0,0158 |
| 19 | Tamer, GS., et al. (2007) | [31] | Aegean | İzmir | 707 | 4 | 0,01 | 0,00 | 0,01 | 1,03 | 0,0028 |
| 20 | Yilmaz, H., et al. (2008) | [7] | East. Anatolia | Van | 2000 | 697 | 0,35 | 0,33 | 0,37 | 1,03 | 0,0107 |
| 21 | Artan, MO., et al. (2008) | [32] | Central Anatolia | Kayseri | 1070 | 55 | 0,05 | 0,04 | 0,07 | 1,03 | 0,0068 |
| 22 | Simsek, Z., et al. (2009) | [33] | Southeastern Anatolia | Şanlıurfa | 71 | 36 | 0,51 | 0,39 | 0,62 | 0,94 | 0,0593 |
| 23 | Çiçek, M., et al. (2012) | [34] | East. Anatolia | Van | 800 | 122 | 0,15 | 0,13 | 0,18 | 1,03 | 0,0127 |
| 24 | Bacalan, F., et al. (2019) | [35] | Southeastern Anatolia | Diyarbakır | 8874 | 1919 | 0,22 | 0,21 | 0,23 | 1,03 | 0,0044 |
| 25 | Turhanoglu, M., et al. (2012) | [36] | Southeastern Anatolia | Diyarbakır | 1079 | 472 | 0,44 | 0,41 | 0,47 | 1,03 | 0,0151 |
| 26 | Eren, C., et al. (2012) | [37] | Southeastern Anatolia | Diyarbakır | 199 | 9 | 0,05 | 0,02 | 0,07 | 1,03 | 0,0147 |
| 27 | Ozdemir, D ., et al. (2005) | [38] | Aegean | İzmir | 76 | 47 | 0,62 | 0,51 | 0,73 | 0,95 | 0,0557 |
| 28 | Çulha, G., et al. (2006) | [39] | Mediterranean | Hatay | 80 | 57 | 0,71 | 0,61 | 0,81 | 0,96 | 0,0506 |

*(Continued)*

| No | First Autor. (Year) | Ref. no | Region | Province | Sample Size | Positive | Prevalence With 95% CI | 95% Lower CI | 95% Upper CI | Weight (%) | Std. err. for Proportion |
|----|---------------------|---------|--------|----------|-------------|----------|------------------------|-------------|-------------|-----------|--------------------------|
| 29 | Aydemir, S., et al. (2024) | [40] | East. Anatolia | Ağrı | 184 | 12 | 0,07 | 0,03 | 0,10 | 1,02 | 0,0182 |
| 30 | Aydin, E., et al. (2022) | [41] | Aegean | Kütahya | 1685 | 127 | 0,08 | 0,06 | 0,09 | 1,03 | 0,0064 |
| 31 | Türk, S., et al. (2012) | [42] | Central Anatolia | Ankara | 39 | 4 | 0,10 | 0,01 | 0,20 | 0,97 | 0,0486 |
| 32 | Ciftci, AO., et al. (1999) | [43] | Central Anatolia | Ankara | 554 | 18 | 0,03 | 0,02 | 0,05 | 1,03 | 0,0075 |
| 33 | Girginkardesler, N., et al. (2003) | [44] | Aegean | Manisa | 400 | 78 | 0,20 | 0,16 | 0,23 | 1,02 | 0,0198 |
| 34 | Cemek, F., et al. (2016) | [45] | East. Anatolia | Van | 71 | 26 | 0,37 | 0,25 | 0,48 | 0,94 | 0,0572 |
| 35 | Cengiz, ZT., et al. (2015) | [46] | East. Anatolia | Van | 1600 | 89 | 0,06 | 0,04 | 0,07 | 1,03 | 0,0057 |
| 36 | Çimen B, and Aktaş O. (2022) | [47] | East. Anatolia | Erzurum | 44884 | 2153 | 0,05 | 0,05 | 0,05 | 1,03 | 0,0010 |
| 37 | Güreser, AS., et al. (2022) | [48] | Central Anatolia | Çorum | 161 | 18 | 0,11 | 0,06 | 0,16 | 1,02 | 0,0248 |
| 38 | Çalik, S., et al. (2011) | [49] | East. Anatolia | Malatya | 1181 | 421 | 0,36 | 0,33 | 0,38 | 1,03 | 0,0139 |
| 39 | Tamer, GS., et al. (2015) | [50] | Marmara | Kocaeli | 145 | 22 | 0,15 | 0,09 | 0,21 | 1,01 | 0,0298 |
| 40 | Okyay, P., et al. (2004) | [51] | Aegean | Aydın | 456 | 145 | 0,32 | 0,28 | 0,36 | 1,02 | 0,0218 |
| 41 | Yentur, DN., et al. (2015) | [52] | Southeastern Anatolia | GAP* | 333 | 149 | 0,45 | 0,39 | 0,50 | 1,01 | 0,0272 |
| 42 | Doğancı, T., et al. (2002) | [53] | Central Anatolia | Ankara | 50 | 3 | 0,06 | 0,00 | 0,13 | 1,00 | 0,0336 |
| 43 | Ulukanlıgil, M., et al. (2003) | [54] | Southeastern Anatolia | Şanlıurfa | 1460 | 906 | 0,62 | 0,60 | 0,65 | 1,03 | 0,0127 |
| 44 | Çeliksöz, A., et al. (2005) | [55] | Central Anatolia | Sivas | 1730 | 599 | 0,35 | 0,32 | 0,37 | 1,03 | 0,0114 |
| 45 | Yentur, DN., et al. (2015) | [56] | Southeastern Anatolia | Şanlıurfa | 100 | 58 | 0,58 | 0,48 | 0,68 | 0,96 | 0,0494 |
| 46 | Balcıoğlu, IC., et al. (2007) | [57] | Aegean | Manisa | 100 | 68 | 0,68 | 0,59 | 0,77 | 0,97 | 0,0466 |
| 47 | Babat, SO., et al. (2018) | [58] | Marmara | İstanbul | 357 | 223 | 0,63 | 0,57 | 0,68 | 1,01 | 0,0256 |
| 48 | Değerli, S., et al. (2009) | [59] | Central Anatolia | Sivas | 2230 | 196 | 0,09 | 0,08 | 0,10 | 1,03 | 0,0060 |
| 49 | Ostan, I., et al. (2007) | [60] | Aegean | Manisa | 294 | 91 | 0,31 | 0,26 | 0,36 | 1,01 | 0,0270 |
| 50 | Atambay, M., et al. (2007) | [61] | East. Anatolia | Malatya | 117 | 53 | 0,45 | 0,36 | 0,54 | 0,97 | 0,0460 |
| 51 | Turhan, E., et al. (2009) | [62] | Mediterranean | Hatay | 177 | 87 | 0,49 | 0,42 | 0,57 | 0,99 | 0,0376 |
| 52 | Çamdalı, S., et al. (2024) | [63] | Central Anatolia | Sivas | 569 | 41 | 0,07 | 0,05 | 0,09 | 1,03 | 0,0108 |
| 53 | Özkan, AM., et al. (2023) | [64] | Central Anatolia | Ankara | 150 | 61 | 0,41 | 0,33 | 0,49 | 0,99 | 0,0401 |
| 54 | Karakuş, İ., et al. (2022) | [65] | East. Anatolia | Iğdır | 400 | 89 | 0,22 | 0,18 | 0,26 | 1,02 | 0,0208 |
| 55 | Beyhan, Y., et al. (2020) | [66] | East. Anatolia | Van | 427 | 12 | 0,03 | 0,01 | 0,04 | 1,03 | 0,0080 |
| 56 | Caner, A., et al. (2020) | [5] | Aegean | İzmir | 82 | 32 | 0,39 | 0,29 | 0,50 | 0,95 | 0,0539 |
| 57 | Taş, ZC., et al.(2019) | [67] | East. Anatolia | Van | 39600 | 20485 | 0,52 | 0,51 | 0,52 | 1,03 | 0,0025 |

*(Continued)*

**Table 1.** (Continued)

| No | First Autor. (Year) | Ref. no | Region | Province | Sample Size | Positive | Prevalence With 95% CI | 95% Lower CI | 95% Upper CI | Weight (%) | Std. err. for Proportion |
|---|---|---|---|---|---|---|---|---|---|---|---|
| 58 | İşler s, et al. (2018) | [68] | East. Anatolia | Van | 105 | 43 | 0,41 | 0,32 | 0,50 | 0,97 | 0,0480 |
| 59 | Maçin, S., et al. (2016) | [69] | Central Anatolia | Ankara | 60 | 12 | 0,20 | 0,10 | 0,30 | 0,96 | 0,0516 |
| 60 | Gökşen, B., et al. (2016) | [70] | Aegean | Manisa | 90 | 11 | 0,12 | 0,06 | 0,19 | 1,00 | 0,0345 |
| 61 | Arıkan, İ., et al. (2016) | [71] | Aegean | Kütahya | 471 | 86 | 0,18 | 0,15 | 0,22 | 1,02 | 0,0178 |
| 62 | Yazgan, S., et al. (2015) | [72] | Central Anatolia | Kayseri | 438 | 44 | 0,10 | 0,07 | 0,13 | 1,03 | 0,0144 |
| 63 | Tüzemen, NÜ., et al. (2014) | [73] | Central Anatolia | Eskişehir | 1049 | 209 | 0,20 | 0,18 | 0,22 | 1,03 | 0,0123 |
| 64 | Değerli, S., et al. (2012) | [74] | Central Anatolia | Sivas | 772 | 155 | 0,20 | 0,17 | 0,23 | 1,03 | 0,0144 |
| 65 | Hamamcı, B., et al. (2011) | [75] | Central Anatolia | Kayseri | 328 | 116 | 0,35 | 0,30 | 0,41 | 1,01 | 0,0264 |
| 66 | Ekinci, B., et al. (2011) | [76] | Aegean | Muğla | 663 | 75 | 0,11 | 0,09 | 0,14 | 1,03 | 0,0123 |
| 67 | Koruk, I., et al. (2010) | [77] | Southeastern Anatolia | Şanlıurfa | 168 | 93 | 0,55 | 0,48 | 0,63 | 0,99 | 0,0384 |
| 68 | Güdücüoğlu, H., et al. (2010) | [78] | East. Anatolia | Van | 195 | 117 | 0,60 | 0,53 | 0,67 | 1,00 | 0,0351 |
| 69 | Köksal, F., et al. (2010) | [79] | Marmara | İstanbul | 29087 | 1242 | 0,04 | 0,04 | 0,05 | 1,03 | 0,0012 |
| 70 | Taş, C., et al. (2009) | [3] | East. Anatolia | Van | 2975 | 1916 | 0,64 | 0,63 | 0,66 | 1,03 | 0,0088 |
| 71 | Taş, C., et al. (2009) | [80] | East. Anatolia | Van | 395 | 114 | 0,29 | 0,24 | 0,33 | 1,02 | 0,0228 |
| 72 | Ataş, AD., et al. (2008) | [81] | Central Anatolia | Yozgat | 367 | 128 | 0,35 | 0,30 | 0,40 | 1,02 | 0,0249 |
| 73 | Karadam, SY., et al. (2008) | [82] | Aegean | Aydın | 133 | 17 | 0,13 | 0,07 | 0,19 | 1,01 | 0,0290 |
| 74 | Yapici, F., et al. (2008) | [83] | Marmara | Kocaeli | 400 | 156 | 0,39 | 0,34 | 0,44 | 1,02 | 0,0244 |
| 75 | Malatyalı, E., et al. (2008) | [84] | Central Anatolia | Sivas | 677 | 251 | 0,37 | 0,33 | 0,41 | 1,02 | 0,0186 |
| 76 | Tamer, GS., et al. (2008) | [85] | Central Anatolia | Sivas | 111 | 37 | 0,33 | 0,25 | 0,42 | 0,98 | 0,0447 |
| 77 | Yılmaz, H., et al. (2007) | [86] | East. Anatolia | Van | 867 | 475 | 0,55 | 0,52 | 0,58 | 1,03 | 0,0169 |
| 78 | Koltas, IS., et al. (2007) | [87] | Mediterranean | Adana,Mersin | 131 | 22 | 0,17 | 0,10 | 0,23 | 1,00 | 0,0327 |
| 79 | Yilmaz, M., et al. (2007) | [88] | East. Anatolia | Elazığ | 448 | 119 | 0,27 | 0,23 | 0,31 | 1,02 | 0,0209 |
| 80 | Otağ, F., et al. (2007) | [89] | Mediterranean | Mersin | 72 | 4 | 0,06 | 0,00 | 0,11 | 1,01 | 0,0270 |
| 81 | Değerli, S., et al. (2006) | [90] | Central Anatolia | Sivas | 264 | 183 | 0,69 | 0,64 | 0,75 | 1,01 | 0,0284 |
| 82 | Alver, O., et al. (2006) | [91] | Marmara | Bursa | 12139 | 525 | 0,04 | 0,04 | 0,05 | 1,03 | 0,0018 |
| 83 | Çelik, T., et al. (2006) | [92] | East. Anatolia | Malatya | 1838 | 415 | 0,23 | 0,21 | 0,25 | 1,03 | 0,0098 |
| 84 | Çeliksöz, A., et al. (2005) | [93] | Central Anatolia | Sivas | 1864 | 694 | 0,37 | 0,35 | 0,39 | 1,03 | 0,0112 |
| 85 | Öztürk, CE., et al. (2004) | [94] | Black Sea | Düzce | 453 | 97 | 0,21 | 0,18 | 0,25 | 1,02 | 0,0193 |
| 86 | Coşkun, S. (1991) | [8] | Southeastern Anatolia | Mardin | 531 | 196 | 0,37 | 0,33 | 0,41 | 1,02 | 0,0209 |
| 87 | Gürses, N., et al. (1991) | [95] | Black Sea | Samsun | 294 | 86 | 0,29 | 0,24 | 0,35 | 1,01 | 0,0265 |
| 88 | Balcı, MK., et al. (1990) | [2] | Central Anatolia | Ankara | 384 | 69 | 0,18 | 0,14 | 0,22 | 1,02 | 0,0196 |

*(Continued)*

**Table 1.** (Continued)

| No | First Autor. (Year) | Ref. no | Region | Province | Sample Size | Positive | Prevalence With 95% CI | 95% Lower CI | 95% Upper CI | Weight (%) | Std. err. for Proportion |
|---|---|---|---|---|---|---|---|---|---|---|---|
| 89 | Öncel, K., (2018) | [96] | Southeastern Anatolia | Şanlıurfa | 2965 | 433 | 0,15 | 0,13 | 0,16 | 1,03 | 0,0065 |
| 90 | Akış, FB., et al. (2018) | [97] | East. Anatolia | Van | 150 | 40 | 0,27 | 0,20 | 0,34 | 1,00 | 0,0361 |
| 91 | Keskinler, D., et al. (1997) | [98] | East. Anatolia | Erzurum | 100 | 59 | 0,59 | 0,49 | 0,69 | 0,96 | 0,0492 |
| 92 | Çelik, T., et al. (2014) | [99] | Southeastern Anatolia | Adıyaman | 450 | 147 | 0,33 | 0,28 | 0,37 | 1,02 | 0,0221 |
| 93 | Çiçek, M., et al. (2011) | [100] | East. Anatolia | Van | 450 | 154 | 0,34 | 0,30 | 0,39 | 1,02 | 0,0224 |
| 94 | Kaplan, M., et al. (2009) | [101] | East. Anatolia | Elazığ | 88 | 33 | 0,38 | 0,27 | 0,48 | 0,96 | 0,0516 |
| 95 | Börekçi, G., et al. (2009) | [102] | Mediterranean | Mersin | 164 | 46 | 0,28 | 0,21 | 0,35 | 1,00 | 0,0351 |
| 96 | Malatyalı, E., et al. (2009) | [103] | Central Anatolia | Sivas | 1449 | 319 | 0,22 | 0,20 | 0,24 | 1,03 | 0,0109 |
| 97 | Kaplan, M., et al. (2009) | [104] | East. Anatolia | Elazığ | 229 | 33 | 0,14 | 0,10 | 0,19 | 1,02 | 0,0232 |
| 98 | Yaman, O., et al. (2010) | [105] | Central Anatolia | Kayseri | 192 | 73 | 0,38 | 0,31 | 0,45 | 1,00 | 0,0350 |
| 99 | Pektaş, B., et al. (2015) | [106] | Central Anatolia | Konya | 2147 | 846 | 0,39 | 0,37 | 0,42 | 1,03 | 0,0105 |

**\*GAP**: Adıyaman, Batman, Diyarbakır, Gaziantep, Kilis, Mardin, Siirt, Şanlıurfa, Şırnak.

followed by Eastern Anatolia (26 studies), then Aegean (16 studies), Southeastern Anatolia (15 studies), Marmara (7 studies), Mediterranean (5 studies) and Black Sea (2 studies) (Fig 2).

The included studies examined a total of 204,754 schoolchildren aged 5–18 years, considered school age, for the presence of IPI. The majority of the included studies used multiple diagnostic methods, including microscopy and serologic and molecular-based methods. A map showing the geographical distribution across the continent based on the included studies is shown in Fig 2.

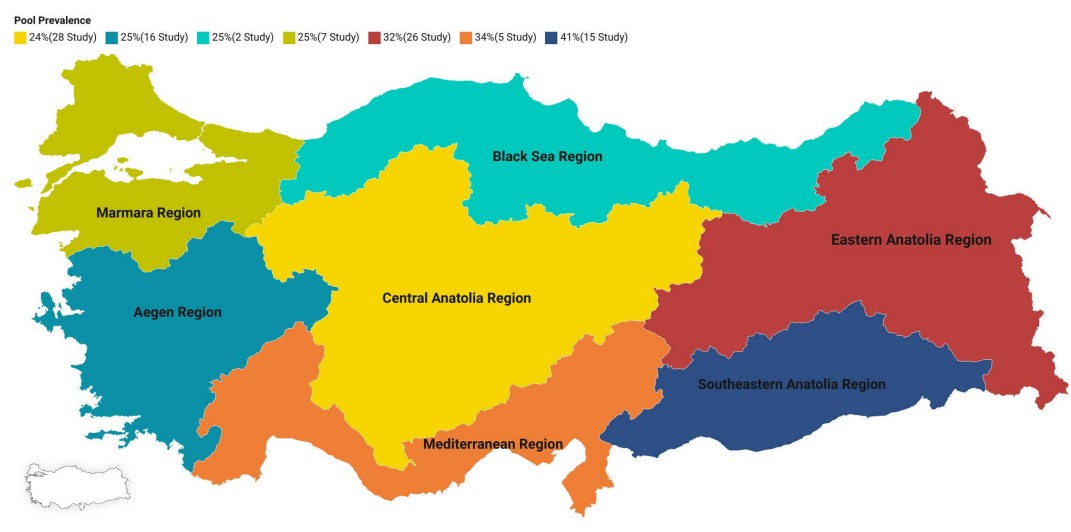

**Fig 2. Distribution of pool prevalences and number of studies by geographical region** https://www.datawrapper.de/_/c8145/.

## Pooled prevalence of intestinal parasite

The prevalence of IPI among school-age children in Turkey was 29% (95% CI: 26%-33%). Significant heterogeneity was observed in all the included studies ($I^2 = 99.88$, $P < 0.001$). Among the regions, Southeastern Anatolia had the highest prevalence at 41% ($I^2 = 99.44$, $P < 0.001$). In the analyses in which parasites were evaluated separately, it was found that *G. intestinalis/duedonalis/labmblia* (11%; 95% CI: 9%-13%, $I^2 = 99.85$) was the most frequently detected parasite in school-age children, followed by *Blastocystis spp., Enterobius vermicularis, Entamoeba coli, Ascaris lumbricoides* and *Entamoeba histolytica/dispar.*

## Quality assessment and publication bias

The funnel plot showed that the publication bias was acceptable. In addition, in this study, there was high heterogeneity ($I^2 = 99.88$, $P < 0.001$), and the Galbraith graph shows the situation more clearly (Fig 3). Begg's rank and Egger's regression intercept tests were used to determine whether the funnel plot asymmetry was larger than expected by chance. Using the findings of the tests by Egger and Begg, we determined that there was no significant publication bias ($P > 0.05$).

## Subgroup analysis

In subgroup analysis, the prevalence in the Aegean region was 25% ($I^2 = 99.57$, $P < 0.001$), in the Black Sea region 25% ($I^2 = 82.50$, $P < 0.001$), in Central Anatolia 24% ($I^2 = 99.29$, $P < 0.001$), in Eastern Anatolia 32% ($I^2 = 99.82$, $P < 0.001$), 25% in the Marmara region ($I^2 = 99.99$, $P < 0.001$), 34% in the Mediterranean region ($I^2 = 98.20$, $P < 0.001$) and finally 41% in the Southeastern Anatolia region ($I^2 = 99.44$, $P < 0.001$). (Fig 4) The results obtained showed that IPI prevalence had a statistically significant effect ($P < 0.001$) (Fig 4).

## Common intestinal parasitic infections among school children

Considering the included studies and the type of parasite detected, the number and percentage of studies belonging to the parasite and the prevalence of the parasites at 95% CI according to the random-effects model, respectively *Chilomastix mesnili* and *Entamoeba hartmanni* (Fig 5), *Cyclospora cayetanensis* and *Trichomonas intestinalis* (Fig 6), *G.intestinalis/duedonalis/labmblia* ((75/99[75.75%]), (11%; 95% CI: 9%-13%, $I^2 = 99.85$)) and *Enterobius vermicularis* ((62/99[62.62%]) (9%; 95% CI: 6%-11%, $I^2 = 99.99$) (Fig 7), *Blastocystis spp.* ((55/99[55.55%]) and (9%; 95% CI: 7%-11%, $I^2 = 99.94$)) and *Entamoeba coli* ((47/99[47.47%]), (6%; 95% CI: 5%-8%, $I^2 = 99.24$)) (Fig 8), Other and *Fasciola hepatica* (Fig 9), *Hymenolepis spp.* and *Ascaris lumbricoides* ((43/99[43.43%]), (6%; 95% CI: 3%-9%, $I^2 = 100.00$)) (Fig 10), *Dientamoeba fragilis, Entamoeba spp.* and *Eteromonas hominis* (Fig 11), *E. histolytica/dispar* ((38/99[38.38%]), (4%; 95% CI: 2%-6%, $I^2 = 99.97$) and *Taenia spp.* (Fig 12), *Iodamoeba butschlii* and *Trichurus trichura* (Fig 13), *Cryptosporidium spp,* and *Endolimax nana* (Fig 14). (Table 2).

## Sensitivity analysis

Sensitivity analyses showed that the exclusion of small studies did not significantly change the summary of the pooled estimates. *Entamoeba spp., Fasciola hepatica, Entamoeba hartmanni, Trichomonas hominis, Retortamonas intestinalis, Strongyloides stercoralis, Dicrocoelium dendriticum* were not excluded because they had a very small effect on the results and this effect was tolerable. The prevalence rate remained within the 95% CI of the corresponding overall prevalence. Overall, the stability of IPI prevalence confirmed the reliability and rationality of our analyses.

## Discussion

Intestinal parasites play a significant role in the prevalence of gastrointestinal diseases in Turkey and globally. These parasites can be transmitted under various conditions and cause severe clinical symptoms, particularly in children. A

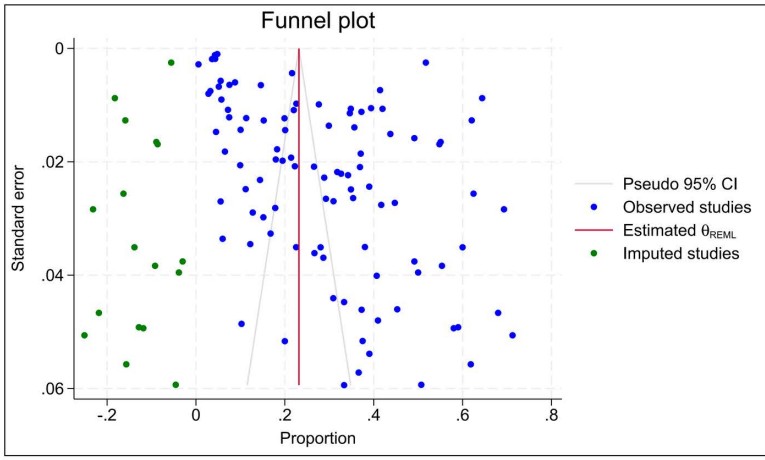

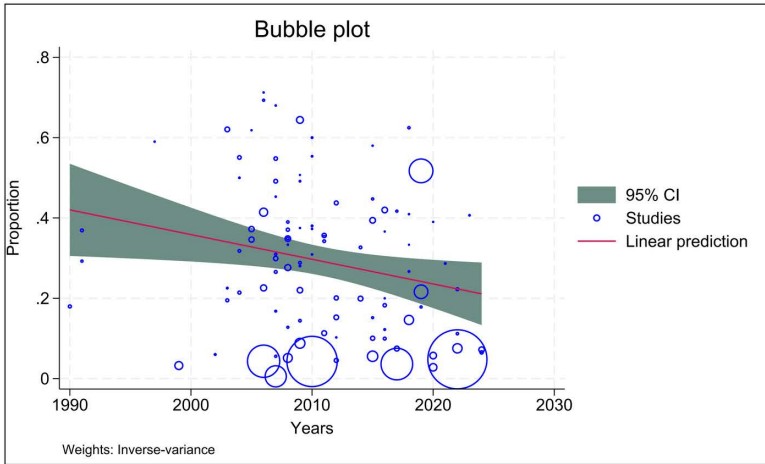

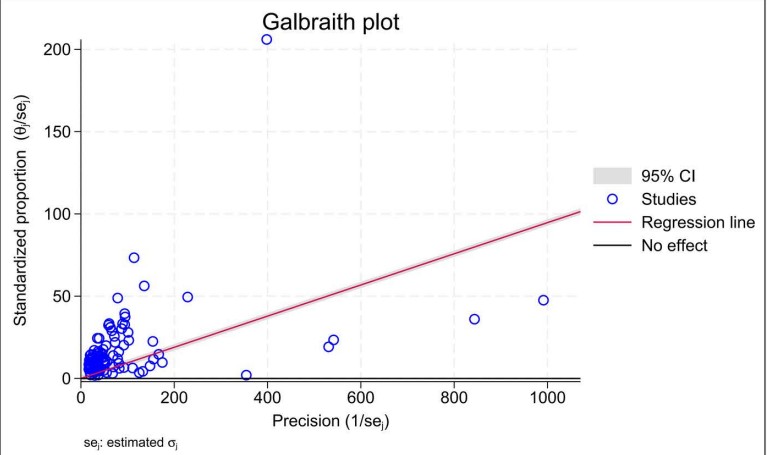

**Fig 3. Funnel plot representing evidence of publication bias, Bubble plot representing of Meta regression and Galbraith plot representing evidence of heterogeneity.**

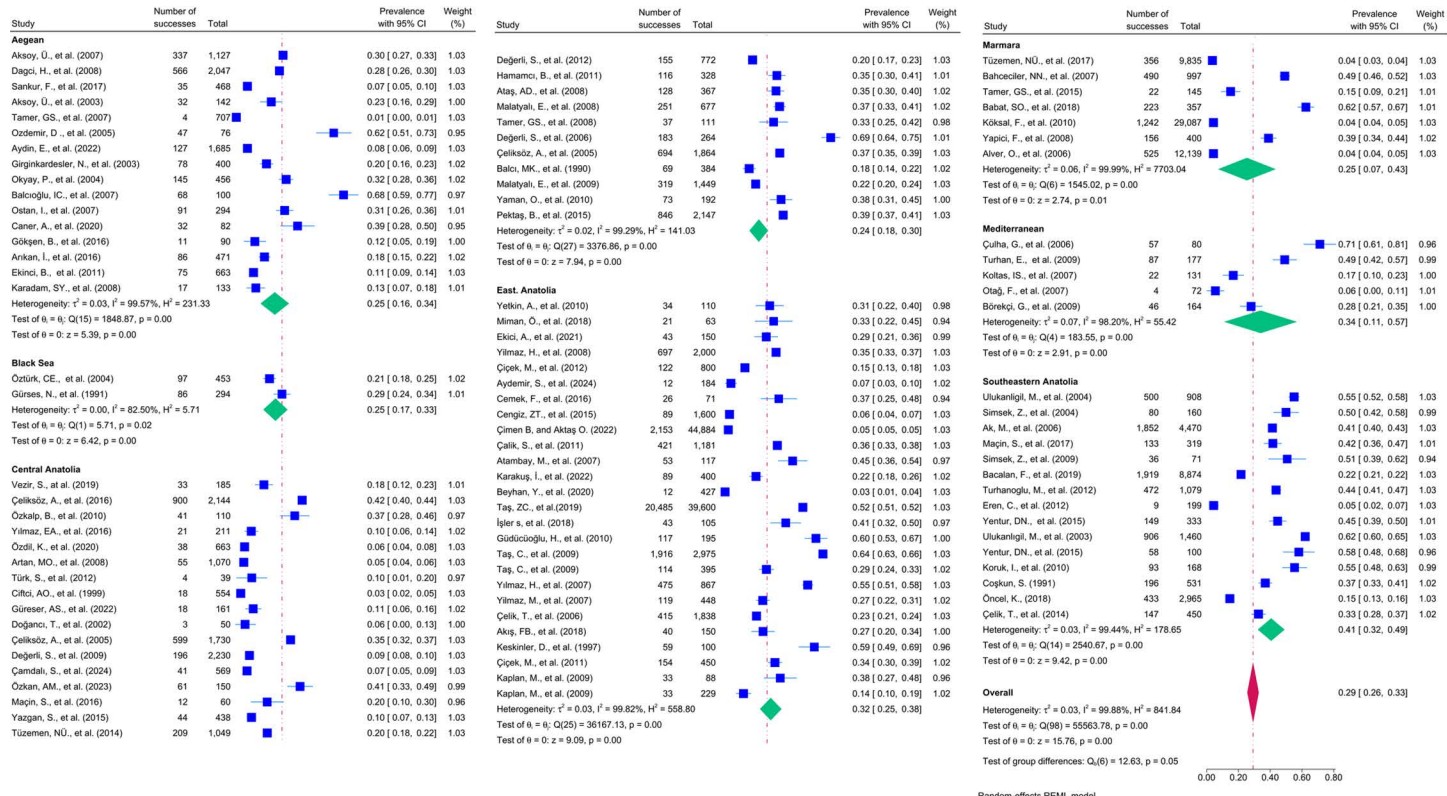

**Fig 4.  Forest plot presenting the prevalence of intestinal parasites in school-age children by region in Turkey (with 95% CI and %weight).**

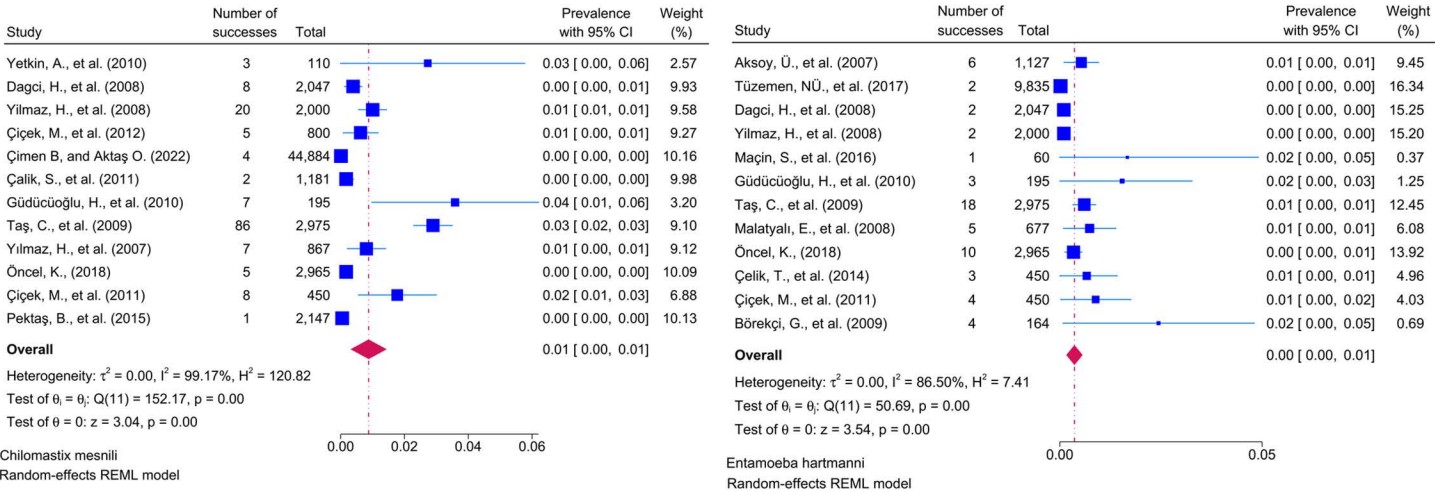

**Fig 5.  Forest plot showing the prevalence of *Chilomastix mesnili* and *Entamoeba hartmanni* in school-age children in Turkey.**

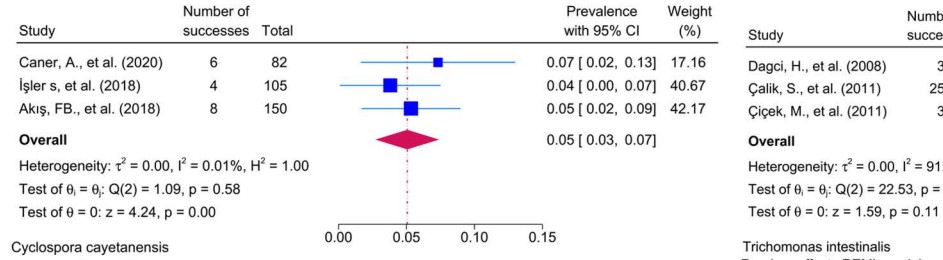
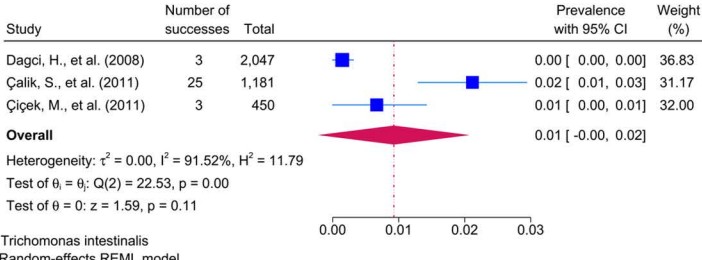

**Fig 6. Forest plot showing the prevalence of *Cyclospora cayetanensis* and *Trichomonas intestinalis* in school-age children in Turkey.**

meta-analysis conducted in Africa reported a prevalence of intestinal protozoa of 25.8% (95% CI: 21.2%-30.3%). In a separate study conducted in Ethiopia, the prevalence was found to be 32.52% (95% CI: 26.24, 38.80). A study in Nepal indicated a prevalence of 55.0% (95% CI: 40.51-69.45). In Iran, the prevalence was 38% (95% CI: 33%–43%). In Colombia, the prevalence was reported to be 55 (95% CI: 48–63). [1,12,107–109].

Meta-regression analysis by year within the scope of the study showed that heterogeneity was at a reasonable level. The most important factor affecting the heterogeneity of the study was that there were no language, region, or year restrictions. This had a positive impact on the reliability and inclusiveness of the study's results. The reasons for the high rates in some regions include the socio-economic structure of the region and differences in education and hygiene habits. The lack of a year-based limitation led to the inclusion of more studies, which had positive effects on the results and reliability. Although more than one diagnostic method was used in the studies included in this study, the gold standard in parasitological examinations is the microscopic detection of the parasite or other forms. In addition, additional tests, such as molecular and serological tests, were among the techniques used to ensure the reliability of the results.

This is the first systematic review and meta-analysis of the prevalence of IPIs among school-age children in Turkey. The current review compiled pertinent data on the prevalence of IPIs from 204.754 school-age children reported in 99 studies conducted in 41 provinces and seven regions of Turkey. The prevalence rates of IPIs in school-age children in Turkey varied to some extent among the included studies.

In this meta-analysis, the prevalence of IPI among school-age children in Turkey was 29% (95% CI: 26%-33%), which is clearly due to poor hygiene, given that the disease agents are transmitted through fecally contaminated food, water, and fingers (Fig 15). When we look at the results of our study, although the prevalence value of intestinal parasites in children in Turkey is lower than the studies conducted in other countries, a rate of 29% is a quantitative indicator that there is a situation that needs to be taken very seriously. To resolve this situation in favor of children, it is of utmost importance that local and public authorities initiate awareness-raising and awareness-raising training on IPI among school-age children and their families and focus on mechanisms to prevent the transmission of parasitic agents.

In particular, in the Southeastern Anatolia Region, where population density is high and crowded large family structures are common, the lack of infrastructure facilities at the desired level is thought to be the reason for the high IPI prevalence. In addition, the fact that average temperatures are favorable for the life cycle of parasites is another factor affecting the high rate of infection in the region. Turkey consists of seven regions and 81 provinces. Although IPIs have been reported in all seven regions, the number of provinces that have not been notified or studied is considerable. In Turkey, with 81 provinces, only 41 provinces reported IPI studies, and these were included in the study. In the light of these data, it should not be ignored that a detailed study covering school-age children in Turkey should be conducted.

Because there is no gold standard test (with 100% accuracy) for the detection of intestinal parasites, different parasitological techniques are used. In most studies included in our review, more than one diagnostic method was used. This increases the reliability of the quantitative data in the present study. The results of the included studies and the parasite

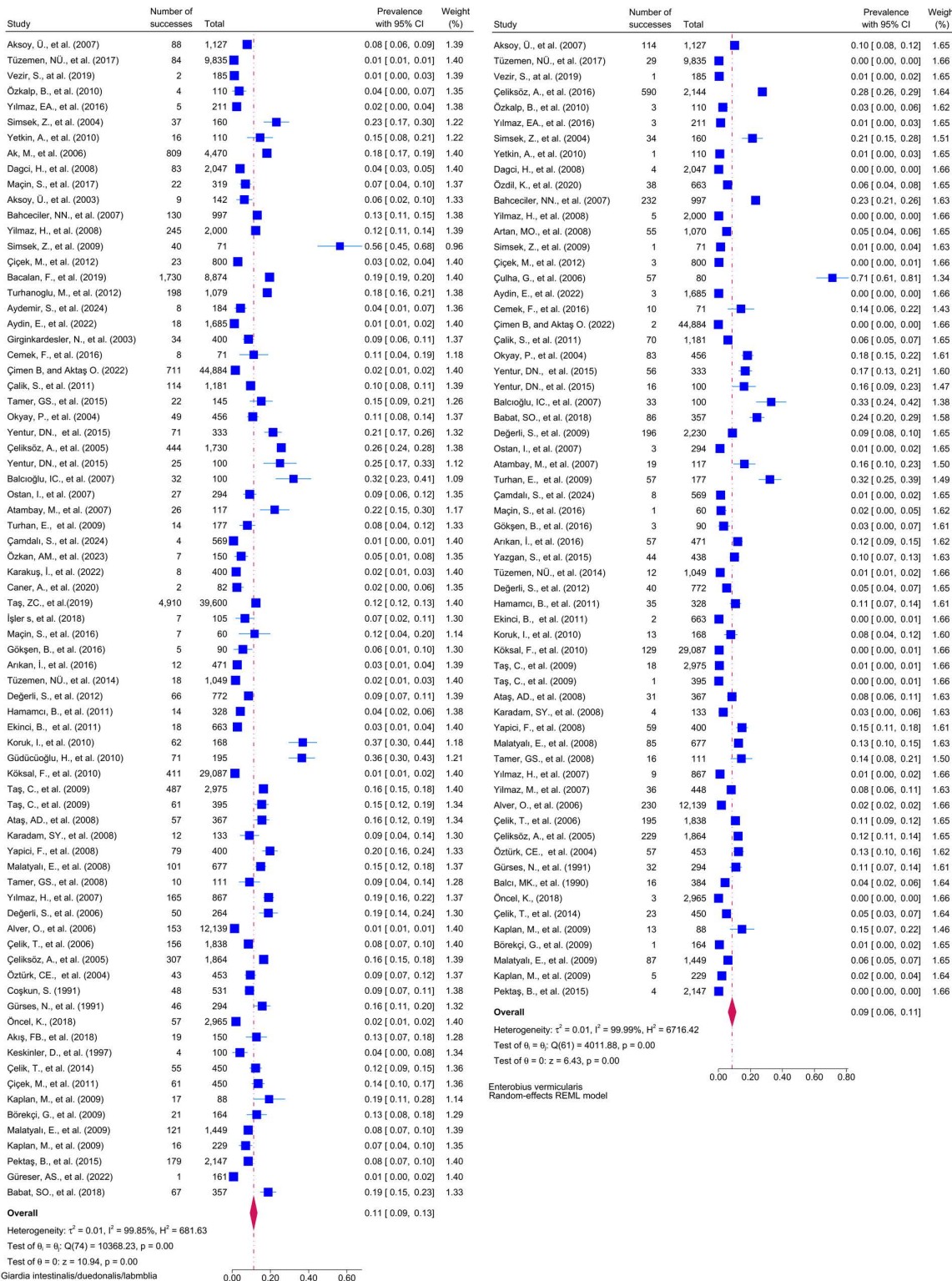

**Fig 7. Forest plot showing the prevalence of *Giardia intestinalis/duedonalis/labmblia* and *Enterobius vermicularis* in school-age children in Turkey.**

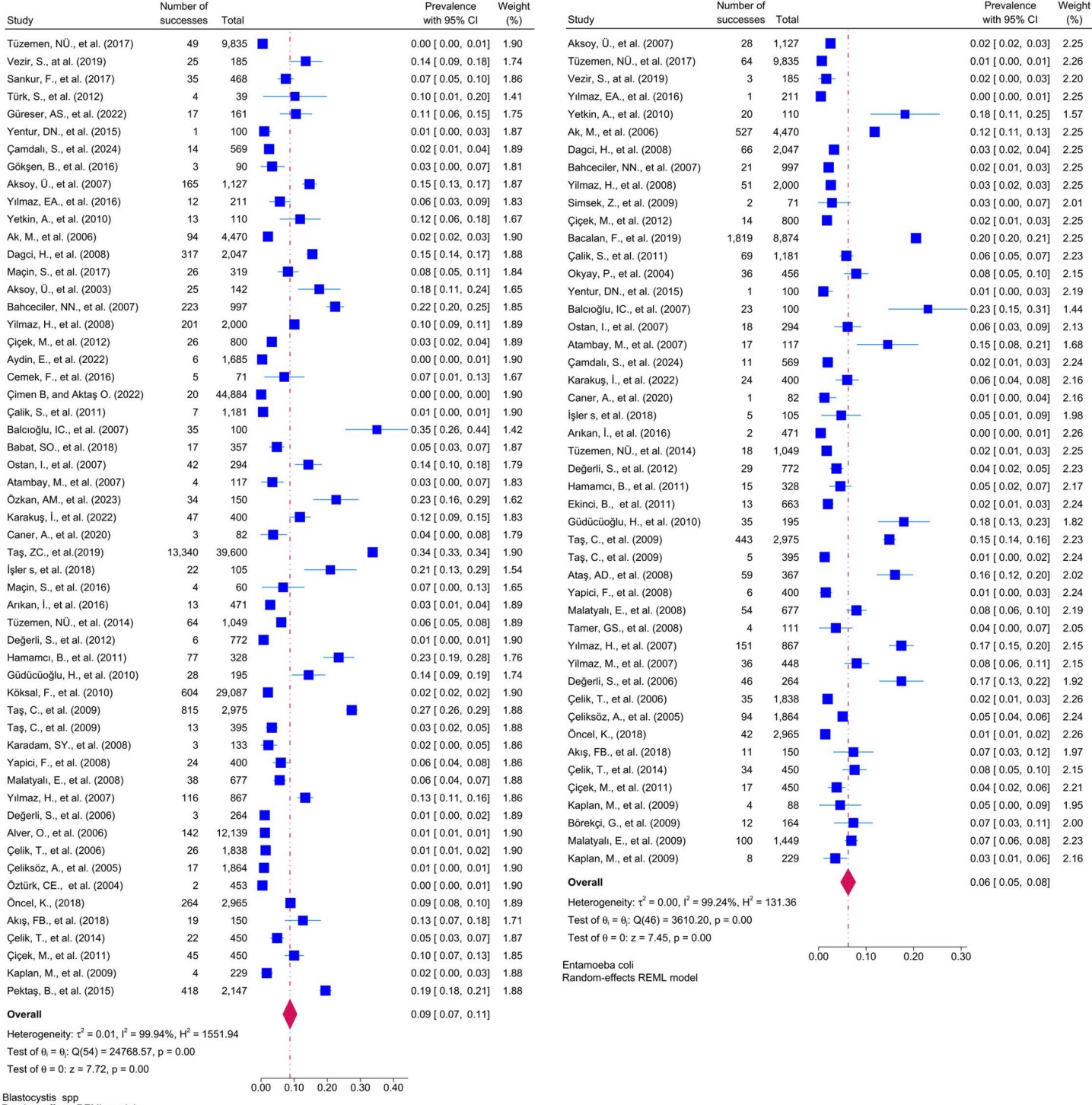

**Fig 8. Forest plot showing the prevalence of *Blastocytis spp.* and *Entamoeba coli* in school-age children in Turkey.**

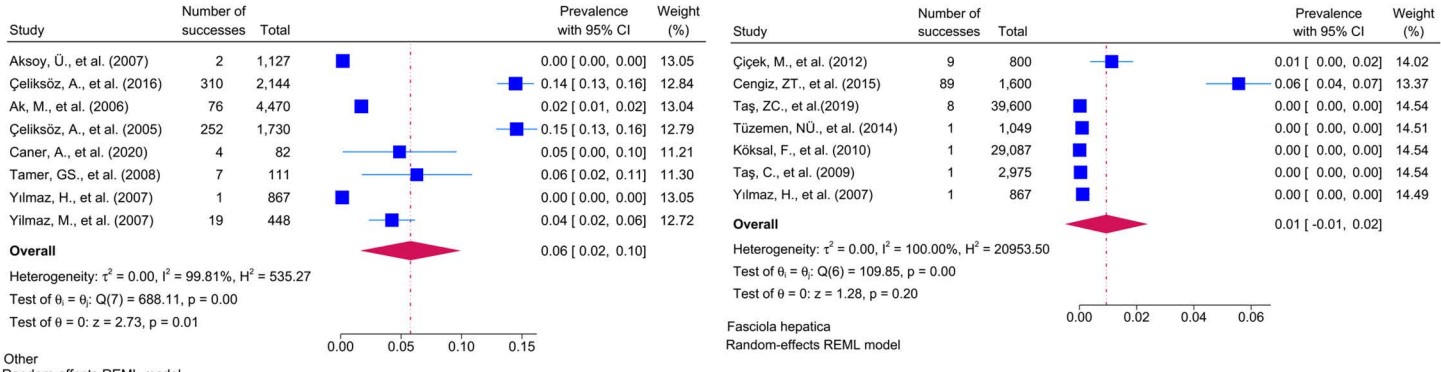

**Fig 9. Forest plot showing the prevalence of Other and *Fasciola hepatica* in school-age children in Turkey.**

species detected are in line with the literature on both parasite type and prevalence. When we look at the literature on parasites belonging to high pool prevalence, the importance of the subject becomes clear.

*Giardia intestinalis/lamblia/duedonalis* is a common protozoan parasite worldwide and causes a disease called giardiasis. The infection is transmitted through contaminated water, food, and poor hygiene. The prevalence rates in developed countries generally range from 1% to 5%. This is often the result of high sanitation levels and access to clean water. However, in these countries, infections can also occur in camp areas, local health problems, or in special risk groups (e.g., immunocompromised individuals) [20,55]. The prevalence rates in developing countries range from 10% to 50%. The high prevalence in these regions is associated with poor sanitation, contaminated water sources, and low hygiene standards.

Giardiasis is more common among children, and school-aged children can play an important role in the spread of infection. Globally, the prevalence is thought to be between 2% and 10%. According to data from the World Health Organization (WHO) and other health organizations, giardiasis has a higher prevalence, especially in low- and middle-income countries [37,47]. This was confirmed in the present study conducted by us. In our study, *the Giardia intestinalis/lamblia/duedonalis* pool prevalence rate was 11%.

*Blastocystis spp.* are protozoan parasites found in the digestive systems of various animals and humans. *Blastocystis spp.* include many different species and can cause a variety of digestive symptoms, such as giardiasis. The prevalence and distribution of infections vary regionally. In countries where clean water and good sanitation systems are in place, the prevalence rates usually range from 10% to 30%. In developing countries, where poor sanitation, contaminated water sources, and low hygiene standards often prevail, prevalence rates can range from 20% to 60%. According to data from the World Health Organization (WHO) and other health organizations, *Blastocystis spp.* infections are more common, particularly in tropical and subtropical regions. The prevalence is estimated to be between 10% and 50%, with a wide range depending on the health conditions and sanitation standards in different regions worldwide [25,42,48]. In our study, the pool prevalence rate of *Blastocystis spp.* was 9%.

*Enterobius vermicularis* is an intestinal parasite (helminth) that is very common worldwide, especially in children and is thought to be associated with both developmental and learning difficulties, with a global rate of 10% to 50% according to literature data. In the present meta-analysis, it was detected at a rate of 9%, which overlaps with the rates reported in the literature. *E. coli* is a common parasitic agent found worldwide, with prevalence rates ranging from 5% to 20%, and is usually transmitted through contaminated water and poor hygiene conditions. E. coli can often be present in humans for many years without any clinical symptoms, and these individuals act as carriers and are instrumental in the spread of the parasite [17,29,32,39]. This parasite, which has a considerable prevalence in Turkey, showed a pool prevalence of 6% in this study (Fig 7).

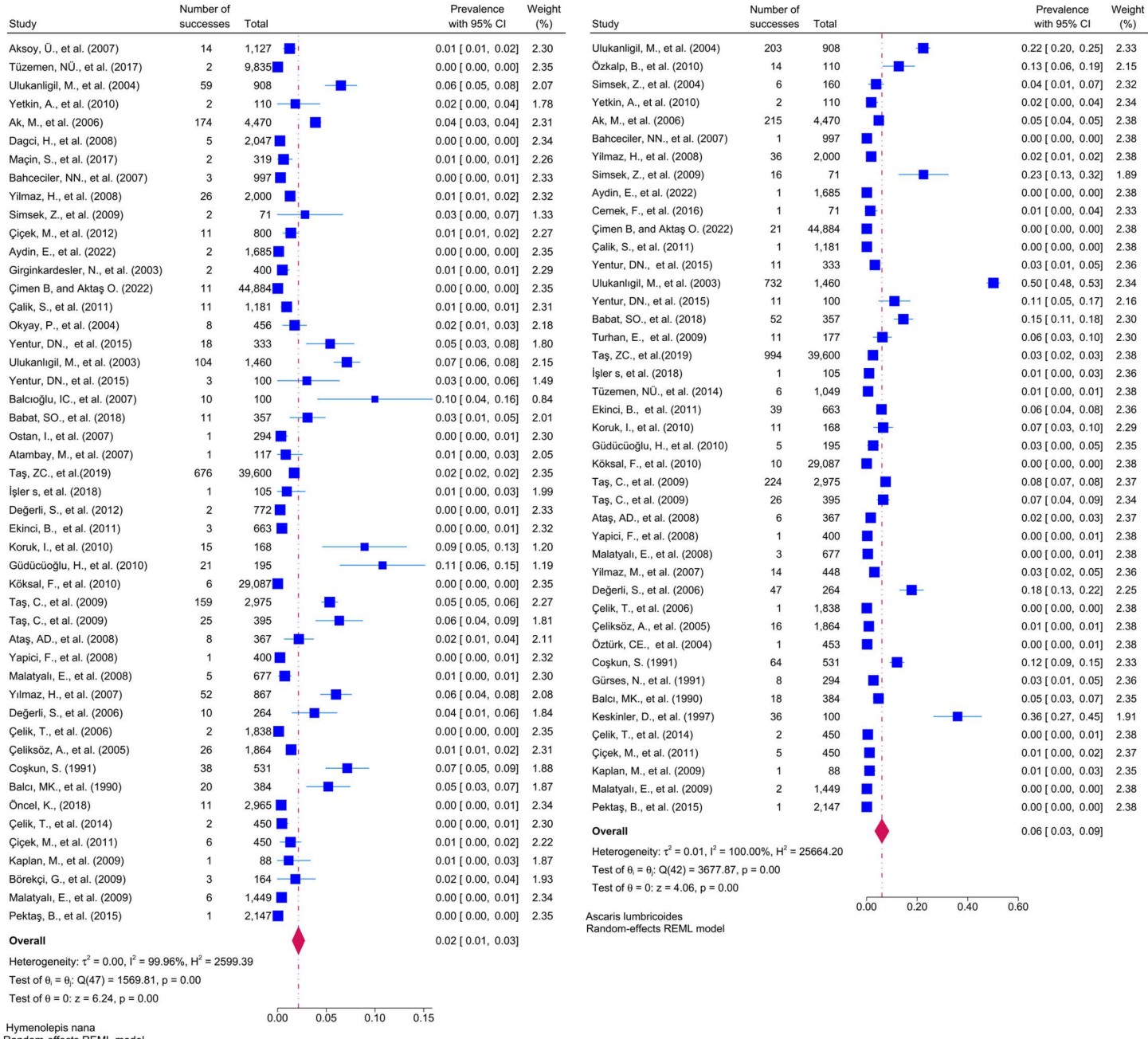

**Fig 10. Forest plot showing the prevalence of *Hymenolepis nana* and *Ascaris lumbricoides* in school-age children in Turkey.**

*Ascaris lumbricoides* is a helminth that is widespread worldwide and causes serious health issues. The eggs of the parasite are excreted in feces and then transmitted to intact buildings through soil, water, food, and hand contamination. After completing its cycle in the human body, the parasite settles in the intestines and continues to live there unnoticed for long periods, causing the person it settles in to become a porter. Globally, the prevalence of this parasite is 10% to 25%, but it can increase to 20% to 80%, especially in tropical and subtropical regions

PLOS Neglected Tropical Diseases

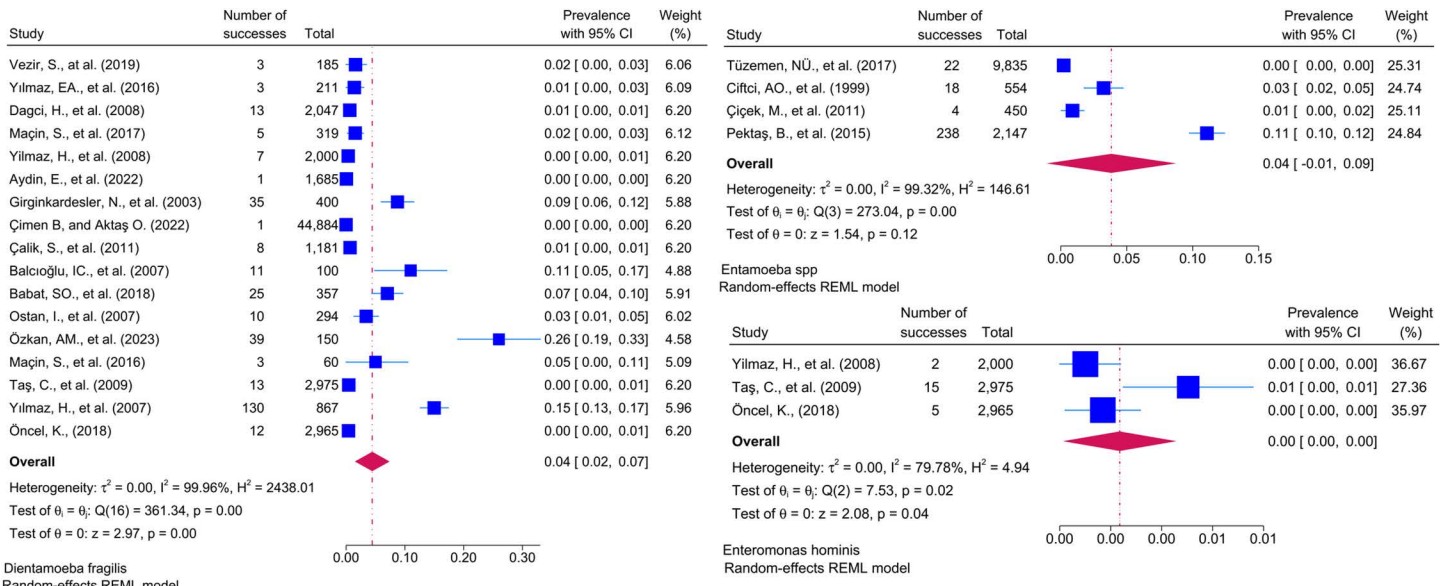

**Fig 11. Forest plot showing the prevalence of *Dientamoeba fragilis*, *Entamoeba spp.* and *Eteromonas hominis*.**

[110–113]. This condition is more common in children. In our study, the pooled prevalence rate of the parasite was 6%.

*Entamoeba histolytica/dispar* is undoubtedly one of a parasitic agent that is an important public health problem worldwide. The disease caused by the parasite is called amebiasis. Depending on the host's immune system, metabolism, and number of ingested parasitic agents, clinical findings can range from mild to severe gastrointestinal disorders [114–116]. In this meta-analysis, *the E. hystolitica/dispar* pooled prevalence was 4%.

Intestinal parasitic infections persist as a critical public health issue, especially in low- and middle-income countries. Various effective interventions have been developed and implemented globally to tackle these infections. These interventions are specifically designed to address local requirements, are supported by public health policies, and are incorporated into sustainable strategies.

The School Health and Community Engagement Program: Endorsed by the World Health Organization (WHO), conducts regular screenings in schools to identify parasitic infections and administer appropriate medications. The program's primary components include health education, hygiene education, and the promotion of awareness among educators [117]. This initiative has notably decreased helminth infections among school-aged children nationwide.

National Anti-helminthic Program: The Ministry of Health in Vietnam has implemented this program, which integrates drinking water and sanitation initiatives with school-based antihelminthic measures. This multifaceted strategy has proven effective not only in managing infections but also in enhancing students' academic outcomes [118].

Integrated Community Health Approach: This initiative, executed in Kenya, incorporated efforts to combat parasitic infections within the framework of community-based health services. Infection transmission was mitigated through home visits, hygiene education, and distribution of anti-parasitic medications by community health workers. The program particularly targets children living in rural areas [119].

Programs Integrated with WASH (Water, Sanitation, and Hygiene): In Brazil, the WASH initiative targeting intestinal parasites seeks to ensure the provision of clean water, adequate hygiene, and proper sanitation infrastructure in educational institutions. Health education campaigns, conducted in conjunction with this initiative, have effectively reduced the risk of infection by enhancing hygiene practices within the community [120].

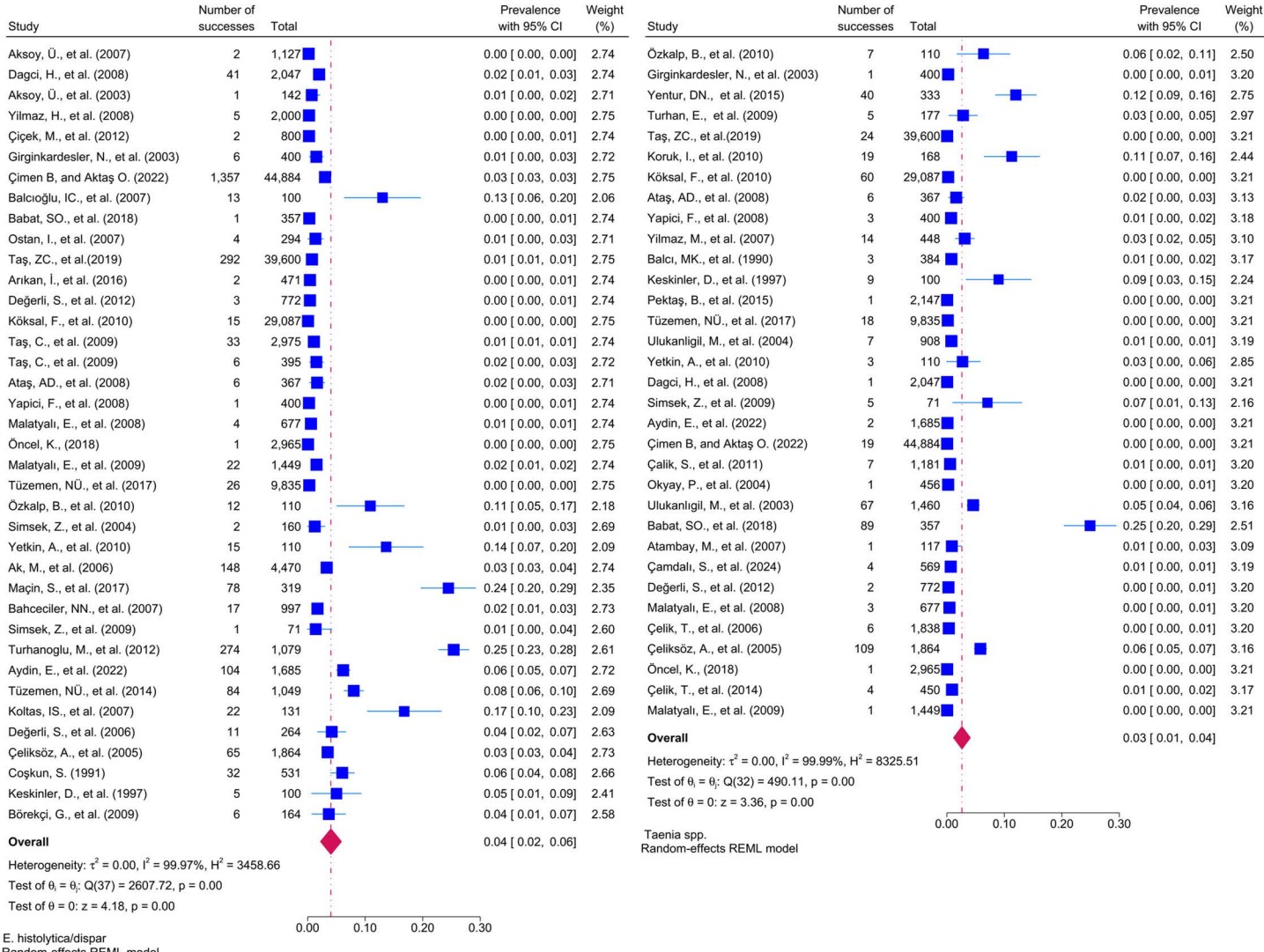

**Fig 12. Forest plot showing the prevalence of *E. histolytica/dispar* and *Taenia spp.* in school-age children in Turkey.**

Hygiene-Based Behavior Change Approaches: In the context of this strategy implemented in Bangladesh, children received training on handwashing, the use of clean water, and food hygiene. This intervention, which emphasized behavioral change and included home visits by health workers, significantly reduced the spread of infections [121].

School health nursing represents: A crucial aspect of public health nursing, with school health nurses assuming essential roles in the prevention of infectious diseases and parasitic infections [122]. Public health nurses are fundamental to the planning, implementation, and evaluation of diagnostic and screening activities, as well as the promotion of personal hygiene, vector control, and management of food, water, and waste [123]. Furthermore, they make significant contributions to health education. This approach has been effectively implemented in various countries globally, resulting in substantial achievements [124].

These instances of successful implementation demonstrate that achieving effective outcomes in addressing intestinal parasitic infections (IPIs) requires not only medical intervention but also comprehensive strategies. These include

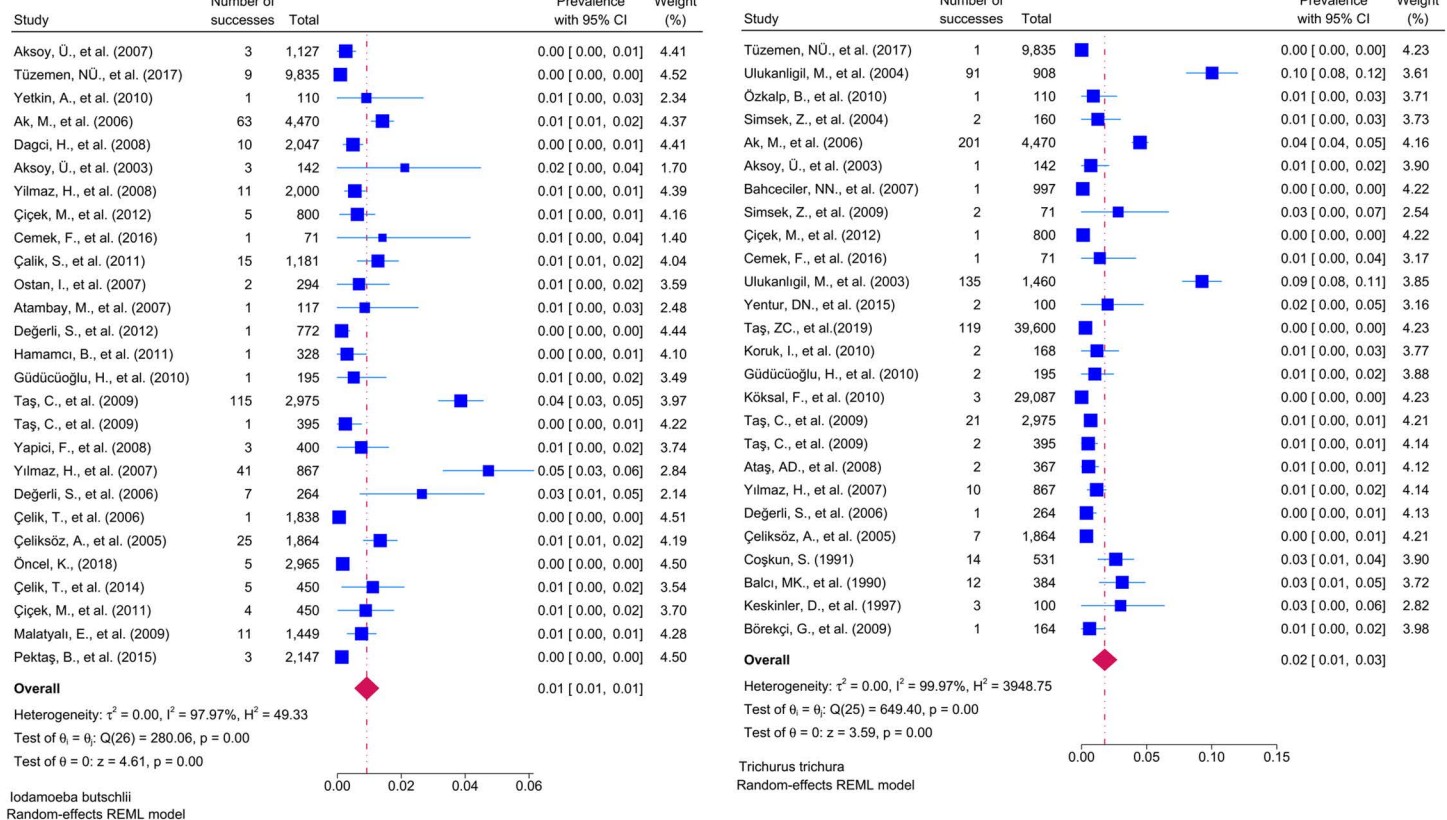

**Fig 13. Forest plot showing the prevalence of *Iodamoeba butschlii* and *Trichurus trichura* in school-age children in Turkey.**

sustainable hygiene and sanitation infrastructure, educational initiatives, behavioral modification strategies, and community involvement. Employing such integrated models in programs implemented in Turkey can significantly reduce the prevalence of intestinal parasites among school-aged children.

To effectively combat intestinal parasites among school-age children in Turkey, the following strategies are recommended.

1. Implementation of National Monitoring and Screening Programs: Regular screening for parasitic infections should be conducted in all regions.

2. Development of Hygiene Education and Behavior Change Programs: Schools and families should be provided hygiene education to enhance awareness and promote behavioral change.

3. Enhancement of Clean Water and Sanitation Infrastructure: Provision of clean drinking water and adequate sanitation facilities is essential, particularly in regions with infrastructural deficiencies, such as the Southeastern Anatolia Region.

4. Expansion of mass drug administration programs: Antiparasitic medications should be distributed in high-risk areas under the coordination of the Ministry of Health.

5. Establishment of a Research Database: A centralized system should be used to collect and analyze regional data on intestinal parasites.

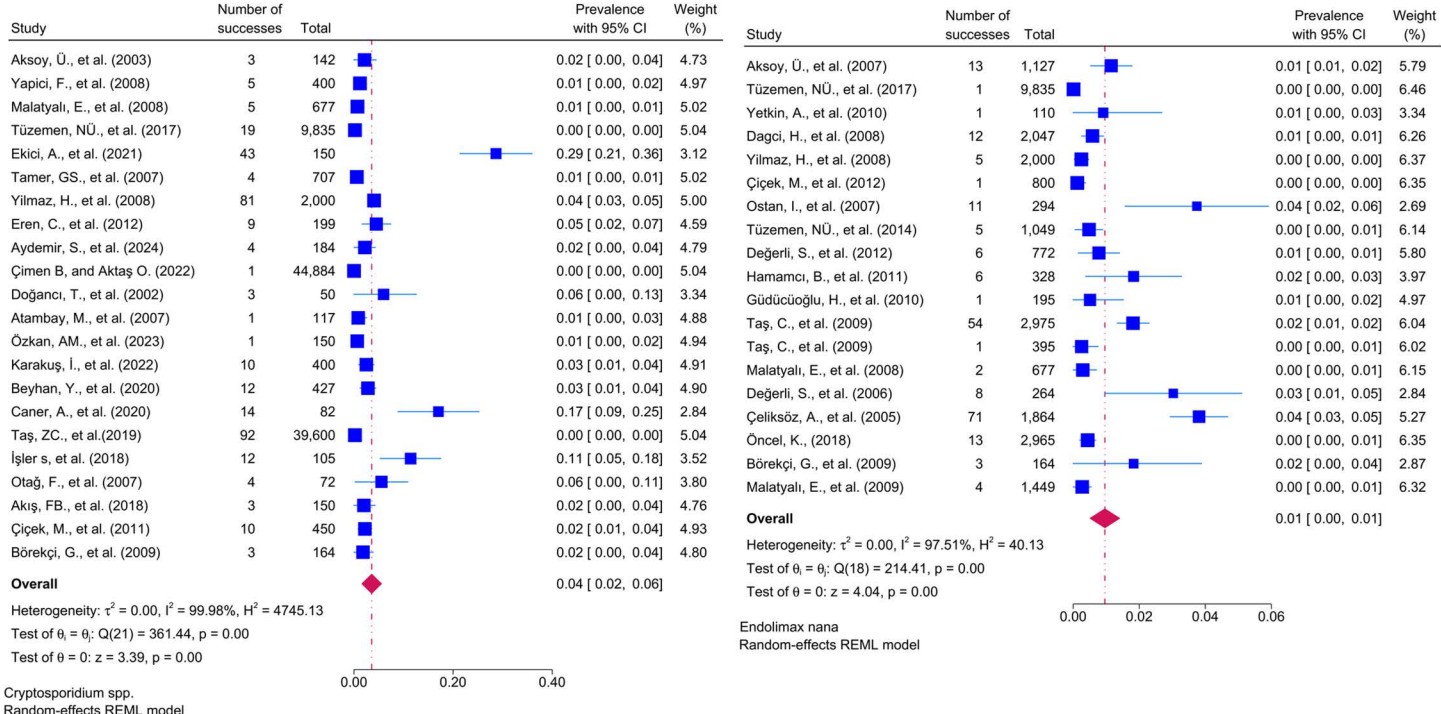

**Fig 14. Forest plot showing the prevalence of *Cryptosporidium spp*. and *Endolimax nana* in school-age children in Turkey.**

6. Collaboration with Local Authorities: Coordinated efforts among municipalities, educational institutions and health directorates are necessary.

7. Intensification of Food Safety Inspections: Food safety inspections in and around schools should be increased, and hygiene standards should be established to ensure food safety.

8. Implementation of School Health Nursing Practices: Structuring school health services under state responsibility, establishing health offices in every school, and employing school health nurses are crucial. Additionally, enhancing communication between nurses, teachers, and parents, utilizing information technologies for efficient communication, and integrating school health services with Community Health Centers should be considered to improve service effectiveness. Given that a significant proportion of parasitic infections, particularly among primary school children, originate from schools, comprehensive health training should be planned for school administrators, teachers, and cleaning staff members.

Based on the literature and data from this meta-analysis, it can be concluded that parasites, which exhibit a high prevalence rate, represent a significant global health concern. Effective infection prevention and control can be achieved through the provision of clean drinking water, implementation of robust sanitation practices, public health education, and promotion of improved personal hygiene. To this end, regional, global planning, and training are of paramount importance.

## Strengths and limitations

The primary strength of this systematic review and meta-analysis lies in its pioneering effort to ascertain the pooled prevalence estimates of intestinal parasitic infections (IPIs) among school-age children across seven regions of Turkey. This

**Table 2. Prevalence of IPIs among school-aged children in Turkey.**

| Parasite name* | Sample Size | Prevalence with 95% CI | 95% lower CI | 95% upper CI |
|---|---|---|---|---|
| G.intestinalis/duedonalis/labmblia | 13.143 | 0,11 | 0,09 | 0,13 |
| Enterobius vermicularis | 3.302 | 0,09 | 0,06 | 0,11 |
| Blastocystis spp. | 17.599 | 0,09 | 0,07 | 0,11 |
| Entamoeba coli | 4.094 | 0,06 | 0,05 | 0,08 |
| Ascaris lumbricoides | 2.876 | 0,06 | 0,03 | 0,09 |
| Other** | 671 | 0,06 | 0,02 | 0,10 |
| Cyclospora cayetanensis | 18 | 0,05 | 0,03 | 0,07 |
| E. histolytica/dispar | 2.719 | 0,04 | 0,02 | 0,06 |
| Dientamoeba fragilis | 319 | 0,04 | 0,02 | 0,07 |
| Taenia spp | 542 | 0,03 | 0,01 | 0,04 |
| Cryptosporidium spp. | 339 | 0,03 | 0,01 | 0,06 |
| Entamoeba spp | 282 | 0,03 | 0,00 | 0,08 |
| Hymenolepis nana | 1.571 | 0,02 | 0,01 | 0,03 |
| Trichurus trichura | 638 | 0,02 | 0,01 | 0,03 |
| Iodamoeba butschlii | 348 | 0,01 | 0,01 | 0,01 |
| Endolimax nana | 218 | 0,01 | 0,00 | 0,01 |
| Chilomastix mesnili | 156 | 0,01 | 0,00 | 0,01 |
| Fasciola hepatica | 110 | 0,01 | -0,01 | 0,02 |
| Trichomonas intestinalis | 31 | 0,01 | 0,00 | 0,02 |
| Enteromonas hominis | 22 | 0,01 | 0,00 | 0,01 |

* Parasite species with zero pool prevalence are not included in the table.

** Parasites used in the articles included in the study, which are very rare in humans and cannot be fully identified.

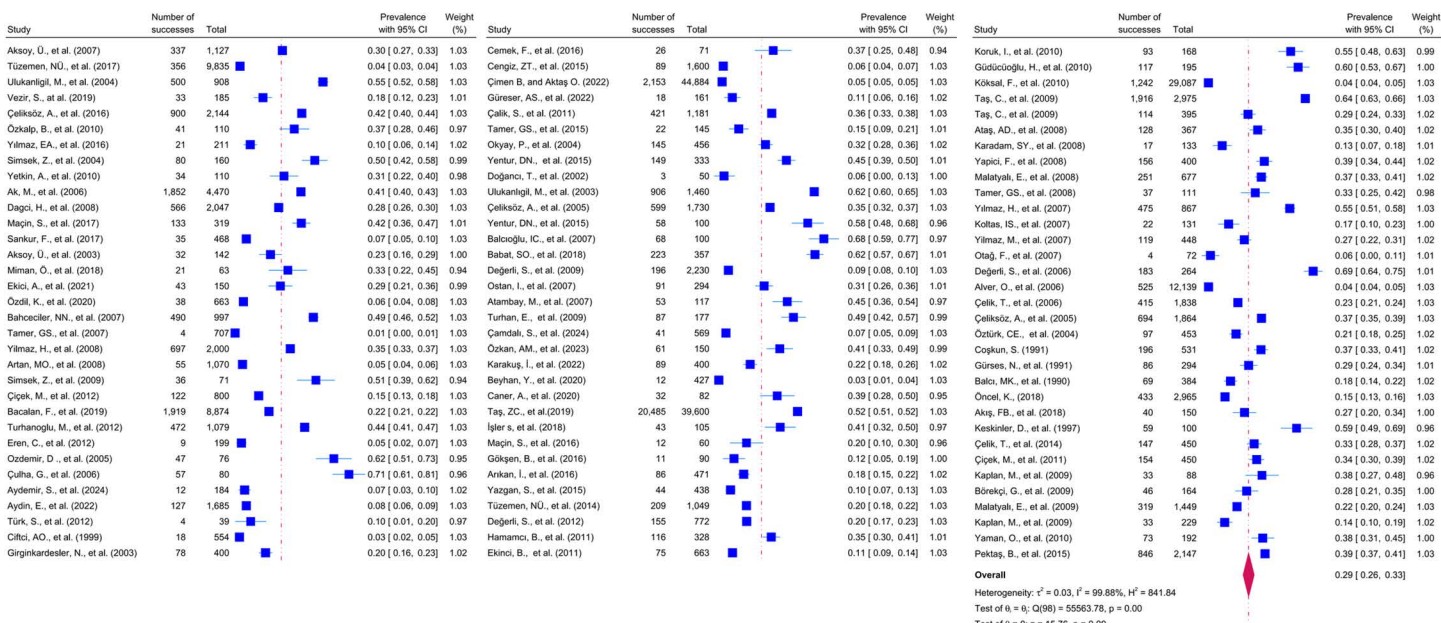

**Fig 15. Overall Forest plot presenting the prevalence of intestinal parasites in school-age children in Turkey (with 95% CI and %weight).**

review is subject to certain limitations. Although data were collected from each region, the sample sizes in some regions were relatively small. Additionally, only 41 of the 81 provinces in Turkey contributed data to the study, resulting in the exclusion of the remaining provinces.

## Conclusion

In Turkey, approximately 29% of school-aged children have been identified as carry one or more species of intestinal parasites in their stool samples. The prevalence is notably higher in Southeast Anatolia, where 41% of children are affected, than in other regions. The predominant parasite detected was *G. intestinalis*, which is associated with gastrointestinal issues and developmental delays in children. This meta-analysis is pivotal in shedding light on the status of intestinal parasitic infections (IPIs) among school-aged children in Turkey, highlighting the ongoing and significant burden these parasites pose. To address this issue, it is essential to focus on poverty alleviation, improve sanitation and hygiene, and implement preventive control measures to curb the spread of intestinal parasites. We propose the formulation of comprehensive action plans to protect children from intestinal parasites throughout Turkey, with a particular focus on Southeast Anatolia.

## Supporting information

**S1 Checklist. PRISMA Checklist.**
(DOCX)

**S1 Table. Search strategies.**
(DOCX)

**S2 Table. Data set used in the study.**
(XLSX)

**S3 Table. Quality assessment of the included studies.**
(DOCX)

## Author contributions

**Conceptualization:** Ahmed Galip Halidi, Kemal Yaran.

**Data curation:** Ahmed Galip Halidi, Kemal Yaran, Selahattin Aydemir, Abdurrahman Ekici.

**Formal analysis:** Ahmed Galip Halidi, Yusuf Dilbilir.

**Investigation:** Ahmed Galip Halidi, Kemal Yaran.

**Methodology:** Ahmed Galip Halidi, Kemal Yaran, Selahattin Aydemir, Abdurrahman Ekici, Yusuf Dilbilir.

**Project administration:** Ahmed Galip Halidi.

**Resources:** Ahmed Galip Halidi, Kemal Yaran.

**Software:** Ahmed Galip Halidi, Yusuf Dilbilir.

**Supervision:** Selahattin Aydemir, Abdurrahman Ekici.

**Validation:** Selahattin Aydemir, Abdurrahman Ekici.

**Visualization:** Ahmed Galip Halidi, Kemal Yaran, Selahattin Aydemir, Abdurrahman Ekici.

**Writing – original draft:** Ahmed Galip Halidi, Kemal Yaran.

**Writing – review & editing:** Ahmed Galip Halidi, Kemal Yaran, Selahattin Aydemir, Abdurrahman Ekici, Yusuf Dilbilir.

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
