## [Decision Letter · Decision Letter 0]

Prevalence of intestinal parasites in school-age children in Turkey: A systematic review and meta-analysis

Dear Dr. HALİDİ,

Thank you for submitting your manuscript to PLOS Neglected Tropical Diseases. After careful consideration, we feel that it has merit but does not fully meet PLOS Neglected Tropical Diseases's publication criteria as it currently stands. Therefore, we invite you to submit a revised version of the manuscript that addresses the points raised during the review process.

Please submit your revised manuscript within 60 days May 20 2025 11:59PM. If you will need more time than this to complete your revisions, please reply to this message or contact the journal office at plosntds@plos.org. Please include the following items when submitting your revised manuscript:

We look forward to receiving your revised manuscript.

Kind regards,

María Victoria Periago

Academic Editor

Guilherme Werneck

Section Editor

Shaden Kamhawi

co-Editor-in-Chief

Paul Brindley

co-Editor-in-Chief

**Journal Requirements:**

At this stage, the following Authors/Authors require contributions: AHMED GALİP HALİDİ, Kemal Yaran, Selahattin Aydemir, Abdurrahman Ekici, and Yusuf Dilbilir. Please ensure that the full contributions of each author are acknowledged in the "Add/Edit/Remove Authors" section of our submission form.

3) Some material included in your submission may be copyrighted. According to PLOSu2019s copyright policy, authors who use figures or other material (e.g., graphics, clipart, maps) from another author or copyright holder must demonstrate or obtain permission to publish this material under the Creative Commons Attribution 4.0 International (CC BY 4.0) License used by PLOS journals. Please closely review the details of PLOSu2019s copyright requirements here: PLOS Licenses and Copyright. If you need to request permissions from a copyright holder, you may use PLOS's Copyright Content Permission form.

Potential Copyright Issues:

- Figures 2. Please provide a direct link to the base layer of the map (i.e., the country or region border shape) and ensure this is also included in the figure legend; and provide a link to the terms of use / license information for the base layer image or shapefile. We cannot publish proprietary or copyrighted maps (e.g. Google Maps, Mapquest) and the terms of use for your map base layer must be compatible with our CC BY 4.0 license.

4) We note that your Data Availability Statement is currently as follows: "All data related to the work during the application is secured in my google drive account. I can share it with the necessary people when requested. In case of a shell of my article, it can be used by PLOS neglected tropical diseases.". Please confirm at this time whether or not your submission contains all raw data required to replicate the results of your study. Authors must share the “minimal data set” for their submission. PLOS defines the minimal data set to consist of the data required to replicate all study findings reported in the article, as well as related metadata and methods (https://journals.plos.org/plosone/s/data-availability#loc-minimal-data-set-definition).

- The points extracted from images for analysis..

**Reviewers' Comments:**

Reviewer's Responses to Questions

**Key Review Criteria Required for Acceptance?**

**Methods:**

-Are the objectives of the study clearly articulated with a clear testable hypothesis stated?

-Is the study design appropriate to address the stated objectives?

-Is the population clearly described and appropriate for the hypothesis being tested?

-Is the sample size sufficient to ensure adequate power to address the hypothesis being tested?

-Were correct statistical analysis used to support conclusions?

-Are there concerns about ethical or regulatory requirements being met?

Reviewer #1: - The objectives are articulated (determine pooled prevalence, identify common parasite species, and compare prevalence across regions). However, the hypothesis is not explicitly stated as a testable proposition and could be framed more explicitly.

- A systematic review and meta-analysis are appropriate for synthesizing prevalence data and comparing regions. The PRISMA guidelines and PROSPERO registration ensure adherence to best practices, yet the PRISMA 2020 statement should be used in place of the outdated 2009 statement.

- While the methodology is sound and robust, a meta-regression addressing the high variability in sample size and study outcomes could shed light on the factors contributing to the observed heterogeneity. This could also allow to test how publication years, diagnostic methods or study settings (urban vs. rural) potentially impact the results.

Reviewer #2: This does not apply.

**Results:**

-Does the analysis presented match the analysis plan?

-Are the results clearly and completely presented?

-Are the figures (Tables, Images) of sufficient quality for clarity?

Reviewer #1: The results align with the analysis plan outlined in the Methods section, including the use of random-effects models, subgroup analyses, and tools such as funnel plots for publication bias. However, the results of the quality assessment as well as additional exploratory analyses, such as meta-regression or sensitivity analyses, could help address the heterogeneity observed and could help understand the factors contributing to regional and methodological variability. While the results are clearly presented, the implications of the aforementioned heterogeneity are not fully addressed. Figures and tables are mostly comprehensive, however there are some issues that need to be addressed (please refer to my editorial suggestions).

Reviewer #2: - It is suggested that the images could be improved in resolution. It is very important to see the detail in the forest graphics, but in the current state this is partly compromised.

- A better description of the quality of the included studies is needed.

**Conclusions:**

-Are the conclusions supported by the data presented?

-Are the limitations of analysis clearly described?

-Do the authors discuss how these data can be helpful to advance our understanding of the topic under study?

-Is public health relevance addressed?

Reviewer #1: The conclusions of this study are well-aligned with the data presented. However, the high heterogeneity across studies raises questions about the reliability of the pooled estimates and regional comparisons. While the manuscript acknowledges heterogeneity, it does not discuss its implications and the authors may consider emphasizing how such variability limits the generalizability and robustness of their conclusions. That said, the authors have identified several important limitations, such as uneven representation of regions and variability in study designs and diagnostic techniques. However, the discussion does not fully address all these points and how these limitations might affect the results.

Overall, the study contributes to a better understanding of intestinal parasite prevalence among school-age children in Turkey, addressing an important area in public health, but the discussion of how these data can inform public health interventions remains limited. Although the authors link the prevalence of intestinal parasites to socioeconomic factors, sanitation, and hygiene, the recommendations remain somewhat broad.

Reviewer #2: The study is important because it focuses on a fragile public and on very prevalent and neglected diseases. However, there might be room for further exploration in the discussion and a more specific conclusion.

I deeply appreciate the significance of the work, and I believe that a more thorough exploration of the approach could further enhance its impact. Other studies demonstrate the efficacy of state and private interventions on a global scale. It may be beneficial to explore alternative approaches that emphasise addressing challenges and celebrating successes.

**Editorial and Data Presentation Modifications?**

Reviewer #1: Time scope of studies

Line 88–95: The manuscript mentions the search date (August 2024) but omits details about the time range for the included studies. It is unclear whether there were any restrictions based on publication year. Please specify the publication years of the included studies (e.g., "Articles published between [year] and [year]"). If no time restrictions were applied, please clarify this explicitly.

Line 155–185 (Characteristics of included studies): Pertaining to the above mentioned point, the omission of the range of years during which the studies were published (or the data of the studies collected) can make it difficult to assess whether the study represents recent trends or includes outdated data. If the prevalence of intestinal parasitic infections is influenced by time-dependent factors (e.g., improvements in sanitation, public health campaigns), it could be interesting to analyze whether prevalence rates have changed over the years.

Data management and study selection

Line 102–103: The text mentions that "attempts have been made to gain missing data or to clarify any uncertainty with corresponding authors" but does not explicitly state whether any articles were excluded due to non-responses from authors and the exclusion criteria include (in line 112-113) "articles that had limited access and those of authors who did not respond to email two times", implying that such articles were excluded if the authors did not respond after two inquiries.The authors might want to include the number of articles excluded due to non-responses or limited access and discuss whether this had any impact on the results (e.g., potential biases introduced by excluding inaccessible studies).

Inclusion and exclusion criteria

Line 109: The third point in the inclusion criteria ends abruptly with "(3)." without providing the criterion. Please clarify this point.

Quality assessment

The manuscript mentions using the Joanna Briggs Institute (JBI) checklist for quality assessment but does not provide results or explain their impact on the study. A summary table or figure (provided with the supplementary documents) showing the quality assessment results (e.g., high, medium, low) for all included studies (or a subset) could improve transparency, strengthen the manuscript and align with best practices in systematic reviews. Potentially, the authors may consider performing a sensitivity analysis to evaluate how the study quality affects the results. If this is not feasible, discuss the potential influence of study quality in the limitations section.

Limitations

Line 357–362: The manuscript mentions that only 41 of 81 provinces were covered but does not discuss the potential consequences this could have for the results of their analyses, e.g., the introduction of regional biases or the impact on the generalizability of the findings.

Similarly, while the authors acknowledge that most studies rely on more than one diagnostic method to detect intestinal parasites, no details are provided on the specific methods used across studies or their potential impact, which could be addressed in the limitations section. More information on diagnostic methods could be included in the supplementary materials.

Data reporting and accuracy

Line 159: The reported numbers for studies in the Marmara region appear to be incorrect. Seven studies should have been reported instead of 17.

Line 214: The authors mention “the free effects model”, which appears to be a typo.

Table and Figure captions

Line 166–167: Figure 1 has two captions (one within the figure and one below it), which is inconsistent.

Line 226–227 (Table 2): While the study explicitly focuses on school-age children (ages 5–18) and all discussions, analyses, and summaries in the text are about school-age children, Table 2, however, references preschool-aged children, which contradicts the rest of the text.

Line 227 (Table 2): The table includes entries like "Other" without clarifying what "Other" entails, making it less informative.

Line 230 & 233: Two figures are labeled as Figure 5; the second should be renumbered as Figure 6.

Language and clarity

Line 154: "Finally, only 99 (14%) of the articles met the eligibility criteria and included in the systematic review and meta-analysis (Fig 1)." → "Finally, only 99 (14%) of the articles met the eligibility criteria and were included in the systematic review and meta-analysis (Figure 1)."

Please revise line 190-192 for clarity and readability: “As a result of the Meta-analysis of the data of the studies included in the study with STATA 18 software, the Funnel plot graph showing the publication bias shows that the publication bias is at an acceptable level.”

Revise table header names in Table 1 (e.g., "Firs autor. (Year)" → "First Author (Year)" and "Samp Size" → "Sample Size").

Address missing punctuation and tab stops, particularly when listing the intestinal parasites (line 116 - 122), but also in other areas of the manuscript, e.g., in line 22 “204754”, in line 274 “Colombia.[1,9,105–107].”, line 310 “individuals)[17,52]”, line 337 “parasite[14,26,29,36]”, line 374 “Anatolia region”.

Reviewer #2: The study needs a complete textual revision. From grammatical correction to detailed revision of terms such as the names of species that represent serious flaws.

**Summary and General Comments**

Reviewer #1: The manuscript titled "Prevalence of intestinal parasites in school-age children in Turkey: A systematic review and meta-analysis" addresses a significant public health issue by quantifying the prevalence of intestinal parasitic infections (IPIs) in school-age children across Turkey. Drawing from 99 articles and 204,754 samples, the study reveals regional heterogeneity in prevalence, with a pooled prevalence estimate of 29%, and identifies common parasitic species, and highlights socioeconomic and geographic factors contributing to the burden of disease.

The study aligns well with the journal's scope and offers important contributions to the understanding of IPIs in Turkey, an important public health issue with implications for targeted interventions and policy-making. However, revisions are necessary to improve the manuscript’s clarity, methodological rigor, and presentation.

Its adherence to standard guidelines, large sample size, and identification of regional disparities and common parasites make the study an important and needed contribution to public health research. However, the study is limited by its use of an outdated PRISMA statement, heterogeneity, limited geographic representation, and insufficient exploratory analyses to address the observed variability. The study is methodologically sound and strong, with the large sample size and multi-regional focus adding robustness to the findings, yet the study could benefit from exploring the observed heterogeneity and thus, an enhanced interpretation of the results. A better articulation of the hypothesis and more detailed discussion (e.g., concerning limitations and implications for public health) may also be considered.

Major revision points:

While the use of PRISMA 2009 demonstrates a commitment to transparency and methodological rigor, the 2020 update incorporates important improvements and provides a more comprehensive framework. To strengthen the manuscript, I recommend the authors revise their methodology and reporting to align with PRISMA 2020. Further, the authors could consider conducting a meta-regression to identify the factors contributing to the observed heterogeneity, such as sample size, publication year, geographic settings (urban vs. rural), study quality and diagnostic methods. Lastly, I suggest to to enhance the discussion of limitations and their implications, as well as providing region-specific public health recommendations.

Reviewer #2: Although the study is nothing new in scientific circles, it represents an important source of information on highly prevalent diseases in a developing country. That alone should be enough to make it relevant and publishable.

However, some errors begin to compromise the quality of the proposal presented and the exploration of the data found in the results can go beyond statistical analysis.

PLOS authors have the option to publish the peer review history of their article (what does this mean? ). If published, this will include your full peer review and any attached files.

**Do you want your identity to be public for this peer review?** For information about this choice, including consent withdrawal, please see our Privacy Policy .

Reviewer #1: No

Reviewer #2: No

**Figure resubmission:**

**Reproducibility:**



---

## [Editor Report · Decision Letter 1]

Response to Reviewers
Revised Manuscript with Track Changes
Manuscript

Shaden Kamhawi

co-Editor-in-Chief

Paul Brindley

co-Editor-in-Chief

**Additional Editor Comments:**

Thanks for the revised version of the manuscript. We acknowledge that most of the reviewers' concerns were adequately tackled. However, there are some minor revisions to be considered:

1) regarding the hypothesis included in the abstract, we have a suggestion for your consideration since the neglected condition is not a hypothesis being tested in the study. Please consider the following suggestion: "The study hypothesizes that the parasite prevalence in school-age children is high, and there is substantial socioeconomic and geographical variation in species-specific prevalence."

2) In the Results section of the abstract, consider changing lines 29/30 to read, "The subgroup analysis revealed that the Southeastern Anatolia is the region with higher prevalence of intestinal parasites among school-age children, with a rate of 41% [...]"

3) Please correct reference 9. It still refers to the old version of PRISMA (2010)

**Reviewers' comments:****Figure resubmission:****Reproducibility:** To enhance the reproducibility of your results, we recommend that authors of applicable studies deposit laboratory protocols in protocols.io, where a protocol can be assigned its own identifier (DOI) such that it can be cited independently in the future. Additionally, PLOS ONE offers an option to publish peer-reviewed clinical study protocols. Read more information on sharing protocols at https://plos.org/protocols?utm_medium=editorial-email&utm_source=authorletters&utm_campaign=protocols

---

## [Editor Report · Decision Letter 2]

Dear Dr HALİDİ,

We are pleased to inform you that your manuscript 'Prevalence of intestinal parasites in school-age children in Turkey: A systematic review and meta-analysis' has been provisionally accepted for publication in PLOS Neglected Tropical Diseases.

Best regards,

María Victoria Periago

Academic Editor

Guilherme Werneck

Section Editor

Shaden Kamhawi

co-Editor-in-Chief

Paul Brindley

co-Editor-in-Chief

---

## [Editor Report · Acceptance letter]

Dear Dr HALİDİ,

We are delighted to inform you that your manuscript, "Prevalence of intestinal parasites in school-age children in Turkey: A systematic review and meta-analysis," has been formally accepted for publication in PLOS Neglected Tropical Diseases.

Best regards,

Shaden Kamhawi

co-Editor-in-Chief

Paul Brindley

co-Editor-in-Chief
